# microRNA arm-imbalance in part from complementary targets mediated decay promotes gastric cancer progression

Zhengyi Zhang[1,2,3], Jingnan Pi[1,2,3], Dongling Zou[4], Xiaoshuang Wang[1,2,3], Jiayue Xu[1,2,3,5], Shan Yu[1,2,3], Ting Zhang[6], Feng Li[7], Xianxie Zhang[1,2,3], Hualu Zhao[1,2,3], Fang Wang[1,2,3], Dong Wang [6]*, Yanni Ma[1,2,3]* & Jia Yu[1,2,3]*

Strand-selection is the final step of microRNA biogenesis in which functional mature miRNAs are generated from one or both arms of precursor. The preference of strand-selection is diverse during development and tissue formation, however, its pathological effect is still unknown. Here we find that two miRNA arms from the same precursor, miR-574-5p and miR-574-3p, are inversely expressed and play exactly opposite roles in gastric cancer progression. Higher-5p with lower-3p expression pattern is significantly correlated with higher TNM stages and poor prognosis of gastric cancer patients. The increase of miR-574-5p/-3p ratio, named miR-574 arm-imbalance is partially due to the dynamic expression of their highly complementary targets in gastric carcinogenesis, moreover, the arm-imbalance of miR-574 is in turn involved and further promotes gastric cancer progression. Our results indicate that miR-574 arm-imbalance contribute to gastric cancer progression and re-modification of the miR-574-targets homeostasis may represent a promising strategy for gastric cancer therapy.

[1] State Key Laboratory of Medical Molecular Biology, Institute of Basic Medical Sciences, Chinese Academy of Medical Sciences (CAMS) & School of Basic Medicine, Peking Union Medical College (PUMC), Beijing 100005, PR China. [2] Key Laboratory of RNA and Hematopoietic Regulation, Chinese Academy of Medical Sciences (CAMS), Beijing 100005, PR China. [3] Department of Biochemistry, Institute of Basic Medical Sciences, Chinese Academy of Medical Sciences (CAMS) & School of Basic Medicine Peking Union Medical College (PUMC), Beijing 100005, PR China. [4] Chongqing Key Laboratory of Translational Research for Cancer Metastasis and Individualized Treatment, Chongqing University Cancer Hospital & Chongqing Cancer Institute & Chongqing Cancer Hospital, Chongqing 400030, PR China. [5] CAS Key Laboratory of Pathogenic Microbiology and Immunology, Institute of Microbiology, Chinese Academy of Sciences, Beijing 100101, PR China. [6] Department of Bioinformatics, School of Basic Medical Sciences, Southern Medical University, Guangzhou 510515, China. [7] Molecular Biology Laboratory, Shanxi Cancer Hospital, Taiyuan 130032, PR China. [8] These authors contributed equally: Zhengyi Zhang, Jingnan Pi, Dongling Zou, Xiaoshuang Wang, Jiayue Xu. *email: wangdong@ems.hrbmu.edu.cn; yanni_ma@126.com; j-yu@ibms.pumc.edu.cn

Gastric cancer (GC) is the fifth leading cause of cancer and the third leading cause of cancer related death which accounts for up to 7% of cancer occurrence and 9% of deaths[1,2]. Although enormous improvements has been achieved for GC diagnosis and treatment, the prognosis of advanced GC remains poor and the overall survival of GC patients is <40% even after a curative surgical resection and adjuvant therapy[3,4]. Therefore, exploring gastric cancer associated genes for early detection or therapeutic targets is essential for improving the prognosis for GC patients.

In last two decades, microRNAs (miRNAs) have been recognized as key players in diverse developmental and cellular processes by regulating gene expression in post-transcriptional level[5–7]. miRNAs are originated from primary miRNA transcripts that fold into hairpins named precursor hairpin, and then are cleaved by two RNases to produce an approximately 22-nucleotide RNA duplex[5,6]. The miRNA duplex is subsequently processed by RISC complex, which unwinds the duplex at the end with weaker hydrogen binding. Next, the strand with free 5′-end is selectively loaded into RISC and served as dominant mature miRNA, while another strand called miRNA* is usually degraded[5,8]. When the miRNA duplex does not contain asymmetric hydrogen-binding ends, both strands can be accumulated in vivo. This process is called arm-sorting or strand selection. Moreover, miRNA and miRNA* preference is not always consistent, it changes depending on the tissue types, development stages, or disease progression[9–12], implying there might be other regulation mechanisms in addition to the hydrogen-binding-based selection rule.

Recently, several studies have proposed that miRNA targets can modulate miRNA strand-selection preference irrespective of hydrogen-binding selection. Target-mediated miRNA protection (TMMP) theory indicates that both release and degradation of miRNA is prevented when its target mRNA is present, and whereby miRNA:miRNA* ratios may be altered depending on target availability[12,13]. Intriguingly, when the target RNA is highly complementary to miRNAs, target RNA can also trigger miRNA degradation by a mechanism involving nucleotide addition and exonucleolytic degradation, which is recognized as target RNA directed miRNA degradation (TDMD). TDMD has been proved to be particularly effective for regulation of miRNAs in neurons[14–17]. However, the contribution of TDMD mediated miRNA decay in altering miRNA arm ratio or selection has not been followed with interest.

It has been reported that the miRNA/miRNA* ratio varies during carcinogenesis. Interestingly, in certain scenarios, one miRNA is favored in tumor tissue and the corresponding miRNA* can be preferred in adjacent normal tissue[11,12]. Yet the intrinsic driver of the miR-5p/miR-3p ratio variation in tumorigenesis and their physiological effects in cancer progression remains unclear. Previously, oncogenic effects of miR-574-5p in colorectal cancer, non-small cell lung cancer and papillary thyroid carcinoma[18–23] as well as tumor suppressive function of miR-574-3p in breast cancer and leukemia had been discovered, respectively[24–27]. But the function of miR-574-5p and miR-574-3p in gastric cancer and the different roles of the two miRNA arms in the same cancer were never described. Here, we report that two miRNA arms from the same precursor, miR-574-5p and miR-574-3p, are reversely expressed in GC patients and play exactly opposite roles in GC progression. More importantly, our results reveal that the inversed change of miR-5p/3p expression in GC is partially due to the dynamic expression of their highly complementary targets through TDMD. The arm-imbalance of miR-574 partially mediated by their targets in turn strongly contribute to and further promote GC progression.

## Results

**miRNA arms show differential expression pattern in GC patients.** Formerly, we investigated the abnormal methylation of miRNA promoter in gastric cancer and accordingly identified a series of aberrantly expressed miRNAs and uncovered their roles in gastric carcinogenesis. They include miR-10a/b[28,29], miR-219-2-3p[30], miR-33b[31], miR-574, and miR-369. Among them, both of the two arms of miR-574, miR-219, and miR-369 can be simultaneously detected in GC tissues. So we further explore the clinical relevance of two arms from the same miRNA precursor in gastric cancer samples, we examined the expression of three pairs of miRNA arms: miR-219-5p/-3p, miR-369-5p/-3p, and miR-574-5p/-3p in GC tissues (Fig. 1a). Intriguingly, miR-574-5p and -3p were reversely expressed whereas two arms of both miR-219 and miR-369 were simultaneously decreased in GC tissues compared with the adjacent non-cancerous tissues. As shown in Fig. 1b, miR-574-5p was upregulated significantly in GC tissues ($p < 0.0001$), while miR-574-3p was downregulated ($p < 0.0001$) compared with normal tissues. Further paired test between GC tissues and the corresponding adjacent non-cancerous tissues indicated two arms of miR-219 and miR-369 were coordinately downregulated in about 50% GC patients, whereas miR-574-5p upregulation accompanied by miR-574-3p downregulation was observed in more than 50% GC patients (Fig. 1c). Notably, miR-574-3p was more prone to be decreased in those GC patients with higher level of miR-574-5p, while increased in GC patients with lower level of miR-574-5p (Fig. 1d), suggesting their reversely differential expression pattern in GC patients.

**miR-574-5p/-3p ratio is correlated with GC progression.** To investigate the correlation of aberrant miR-574-5p/-3p expression with GC progression, the expression level of miR-574-5p/-3p was statistically analyzed with the clinical pathological characteristics of GC patients (Fig. 1e, Supplementary Tables 1 and 2). A more aggressive tumor stage (stage III/IV versus I/II) was correlated with higher level of miR-574-5p ($p = 0.0028$) and lower level of miR-574-3p ($p < 0.0001$) (Fig. 1f, Supplementary Fig. 1). No significant correlation was observed in age, gender, invasion, location, and classification. To further explore the correlation of reversely expressed miR-574-5p/-3p with GC progression, we defined a risk factor (RF) as miR-574-5p upregulation or miR-574-3p downregulation in GC tissues and sorted GC patients into three groups (No. RF = 0, 1, 2). It was obvious that the percentage of III + IV stage patients in No. RF = 2 group was dramatically higher than that in the other two groups (Fig. 1g), indicating that GC patients with higher miR-574-5p level accompanied by lower miR-574-3p level had more aggressive tumor phenotype than others and the inversely expression of miR-574-5p/-3p might play important roles in GC progression. In addition, Kaplan–Meier survival analysis suggested that GC patients with concurrent upregulated miR-574-5p and downregulated miR-574-3p expression were correlated with poorer survival (Fig. 1h, Supplementary Table 3).

**miR-574-5p and -3p divergently control GC cell growth.** To determine whether miR-574-5p/-3p played opposite role in gastric carcinogenesis, we overexpressed these miRNA arms in two GC cell lines (MGC-803 and HGC-27) to evaluate their effect on cell proliferation and apoptosis, respectively. Cell growth curve showed that miR-574-5p significantly triggered cell proliferation of both MGC-803 and HGC-27 cells, while miR-574-3p suppressed their proliferation (Fig. 2a). Meanwhile, miR-574-5p markedly improved the colony formation ability of MGC-803 and HGC-27 but miR-574-3p inhibited such capacity contrarily (Fig. 2a). In addition, the percentage of

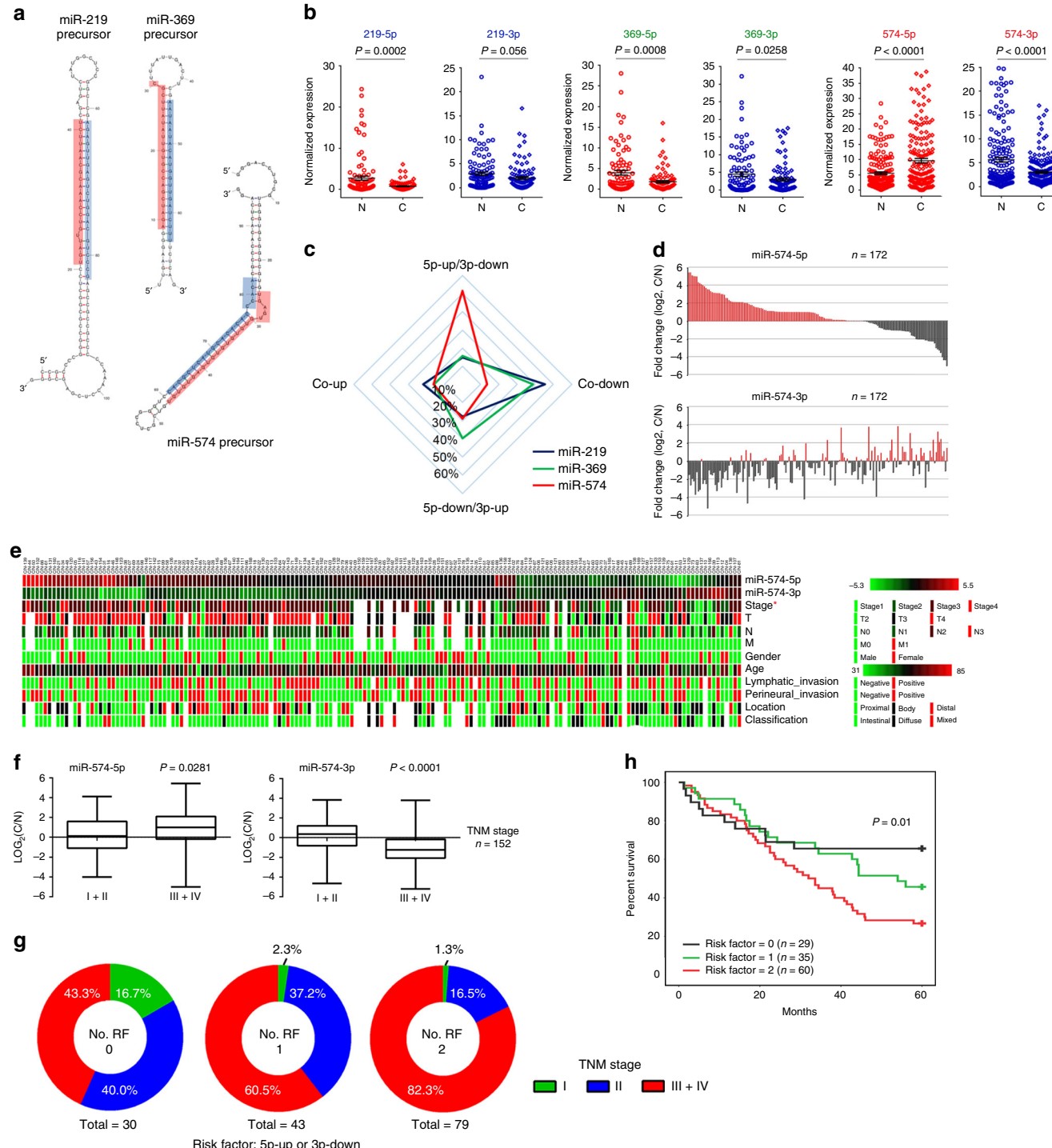

**Fig. 1** miR-574-5p and miR-574-3p were reversely expressed in GC patients. **a** The predicted secondary structure of miR-219, miR-369 and miR-574 precursors. **b**, **c** The expression changes of three pairs of miRNA arms: miR-219-5p/-3p, miR-369-5p/-3p, and miR-574-5p/-3p in GC tissues. **d** The reversed expression of miR-574-5p/-3p in GC patients. **e** Correlation of miR-574-5p/-3p expression with clinical pathological characteristics of GC patients. **f**, **g** Correlation of miR-574-5p/-3p expression with TNM stages of GC patients. The center line indicates the median expression level, bounds of box indicates the interquartile range and whiskers indicates the maximum and minimum value (**f**). A risk factor was defined as miR-574-5p upregulation or miR-574-3p downregulation (**g**). **h** Kaplan–Meier survival analysis showed the correlation of arm-imbalance of miR-574 with survival rate of GC patients. Data are shown as means ± s.d.

early, but not late, apoptotic cells was clearly decreased in miR-574-5p overexpressed cells, while increased in miR-574-3p overexpressed cells (Fig. 2a, Supplementary Fig. 1A). To validate these findings in vivo, MGC-803 cells were inoculated subcutaneously in posterior flanks of nude mice. When tumors reached 50 mm³, synthetic miR-574-5p, -3p and scrambled oligonucleotides were injected directly into the tumors every 6 days for five times, respectively (Supplementary Fig. 1B). In line with the in vitro result, miR-574-5p accelerated the growth rate of MGC-803-engrafted tumors compared with scrambled

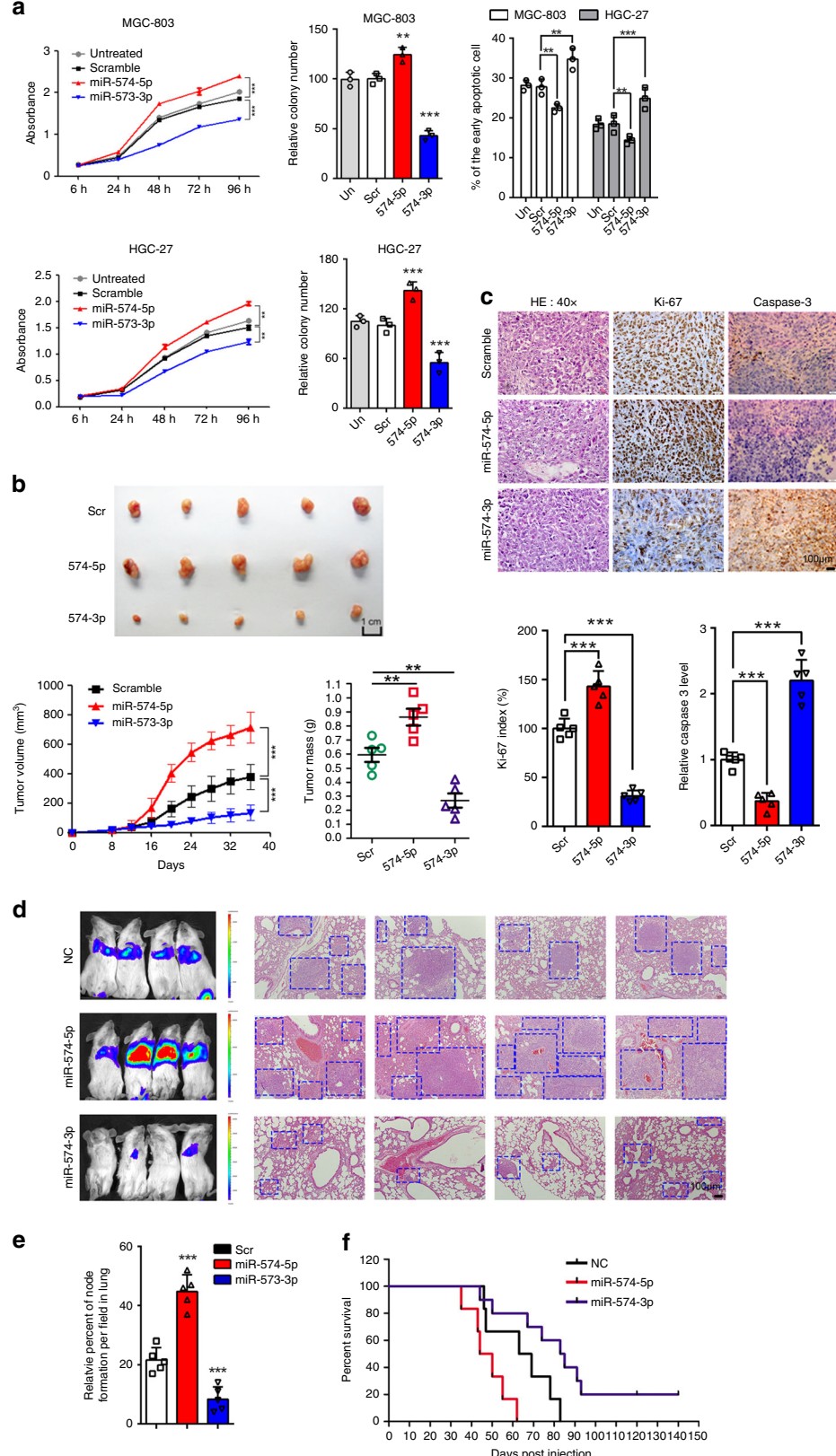

control, while miR-574-3p inhibited the growth of engrafted tumors (Fig. 2b, Supplementary Fig. 1C). The volumes and weights of tumors treated by miR-574-5p mimic were significantly increased than scrambled controls, whereas those of miR-574-3p mimic treatment resulted in suppressed

tumor growth (Fig. 2b). In miR-574-5p mimic treated tumors, proliferation marker Ki-67 was upregulated and apoptotic marker caspase-3 was downregulated. The opposite result was observed in miR-574-3p group (Fig. 2c, Supplementary Fig. 1D).

**Fig. 2** miR-574-5p and -3p divergently control GC cell growth and metastasis. **a** The effects of miR-574-5p/-3p overexpression on GC cell (MGC-803 and HGC-27) proliferation (left), colony formation ability (middle), and early apoptosis (right). Three technical replicates from a single experiment representative of three independent experiments. **b** Photographs of MGC-803-engrafted tumors treated with scramble or miR-574-5p/-3p mimics at the end of the experiment (day 40), and tumor volumes as well as tumor weight are shown at the indicated days ($n = 4$). **c** Immunohistochemistry analysis of Ki-67 and caspase-3 expression in tumors from xenograft mice. **d** Bioluminescence imaging of mice showed GC cell metastasis in vivo and H&E staining of the lung tissues. **e** The relative percent of metastatic nodules formation in lung. **f** The survival rate of mice transplanted with MGC-803 cells treated with scramble or miR-574-5p/-3p mimics. Data are shown as means ± s.d. *$p < 0.05$, **$p < 0.01$, ***$p < 0.001$, Student's $t$-test.

As the importance of metastasis in cancer progression, we next investigated the role of miR-574-5p/-3p in GC cell movement in vitro and in vivo. The wound-healing and invasion assays were performed to assess GC cell migration and invasion. miR-574-5p mimic treated MGC-803 cells completely sealed linear scratch wounds at 36 h after injury, scramble control treated MGC-803 cells sealed about 80% of the wound area after 36 h, whereas miR-574-3p mimic treated cells sealed only about 40% of the wound area. Similar results were observed in HGC-27 cells (Supplementary Fig. 1F). Meanwhile, miR-574-5p can also significantly increased the number of invasive GC cells passing through Matrigel-coated membrane matrix compared with scramble, whereas miR-574-3p did the opposite (Supplementary Fig. 1E). To confirm these findings, we conducted in vivo metastasis assays. In doing so, $5 \times 10^5$ viable MGC-803 cells stable expressing luciferase transfected with miR-574-5p/-3p or scramble control were injected into the lateral tail veins of nude mice. Five weeks after injection, bioluminescence imaging of mice showed miR-574-5p promoted metastasis of GC cells to lung, and miR-574-3p obviously impaired metastasis of the cells (Fig. 2d). The animals were sacrificed and the lungs were dissected for microscopic histology. The number of lung metastases in mice injected with miR-574-5p mimic treated cells was significantly more than that with scramble control, while that with miR-574-3p-treated cells was seriously less than control. Hematoxylin and eosin (H&E) staining was also performed to assess the pathologic properties of the lung tissues, which showed the most metastatic nodules in miR-574-5p-treated group and the least in miR-574-3p-treated group (Fig. 2d, e). Importantly, a dramatic reduction in survival of animals transplanted with miR-574-5p-treated cells was observed, in contrast, miR-574-3p prolonged survival (Fig. 2f).

**miR-574-5p/3p target endogenous QKI6 and ACVR1B, respectively**. Although miR-574-5p and -3p originated from the same miRNA precursor, they oppositely acted in GC progression evidenced by the above findings. We speculated miR-574-5p/-3p exerted contrary roles through targeting different RNA targets, mediated by their different seed sequences. Traditionally, we interrogated the TargetScan and miRanda miRNA target prediction programs to identify targets of miR-574-5p/3p. QKI6 and ACVR1B containing binding sites for miR-574-5p/3p were identified as possible target sites of miR-574-5p/3p, respectively (Fig. 3a). Luciferase reporter assay revealed miR-574-5p repressed the luciferase activity of QKI6 3′ UTR reporter, while mutation of miR-574-5p binding sites abrogated this reduction. In the same way, miR-574-3p inhibited the luciferase activity of ACVR1B 3′ UTR reporter and the inhibition was dependent on the miR-574-3p binding sites (Fig. 3b). Furthermore, elevating miR-574-5p or -3p in MGC-803 or HGC-27 cells reduced the QKI6/ACVR1B mRNA and protein level (Fig. 3c, Supplementary Fig. 2A). These findings indicated that QKI6/ACVR1B were direct targets of miR-574-5p/-3p, respectively, and could be negatively regulated by them in GC cells. QKI was a general tumor suppressor in a variety of human cancers by controlling gene expression post-transcriptionally, including regulating RNA alternative splicing, RNA stability, RNA processing etc[23,32–37]. miR-574-5p has been

shown to target QKI6 to increase proliferation, migration and invasion in human colorectal cancer[33]. To determine whether miR-574-5p's oncogenic role in GC is dependent on QKI6 targeting, we simultaneously overexpressed QKI6 to restore its repression induced by elevated miR-574-5p level in both MGC-803 and HGC-27 cells (Fig. 3d). Consistent with the restored expression of QKI6 protein, promotion of GC cell proliferation and invasion by miR-574-5p mimics was rescued by the addition of QKI6 (Fig. 3e, f, Supplementary Fig. 2B, C). These data confirmed the oncogenic role of miR-574-5p in gastric carcinogenesis was partially through targeting QKI6. In a similar way, we also proved inhibition of GC cell proliferation and invasion by miR-574-3p was partially through targeting oncogenic ACVR1B[38,39] (Fig. 3e, f, Supplementary Fig. 2B, C).

Further, we examined QKI6 and ACVR1B expression in 12 pairs of GC tissues, demonstrating increased miR-574-5p and decreased miR-574-3p levels. QKI6 was clearly downregulated, while ACVR1B was upregulated in these GC tissues comparing to the adjacent non-cancerous tissues (Fig. 3g, Supplementary Fig. 2D), indicating an inverse correlation between miR-574-5p/-3p and their targets.

To investigate the global molecular changes induced by miR-574-5p/-3p, we overexpressed miR-574-5p/-3p in MGC-803 and performed RNA deep sequencing, respectively. There were 4786 genes showing significantly differential expression in miR-574-5p mimic treated GC cells (Supplementary Fig. 2E, Supplementary Data 1). Among them, many known oncogenes, such as FRAT1, FGFR2, CCND1, BCL2, MYC, JAK2 etc. were clearly increased, while some known tumor suppressor genes, such as PTEN, RB2, RHOB, TET2, ING1 were decreased (Fig. 3h). Furthermore, as compared with control, enforced expression miR-574-5p in GC cells highly enriched several terms reported to improve cancer cell growth, including positive regulation of cell division and cell cycle, ribosomal biogenesis and DNA replication, meanwhile showed dramatic downregulation of the terms involved in tumor suppression, including cell cycle arrest, negative regulation of cell migration, negative regulation of cell proliferation, negative regulation of ERK1 and ERK2 cascade (Fig. 3i). Similarly, many genes were downregulated in miR-574-3p overexpressed GC cells, including many known oncogenes, such as AKT1, AKT2, EGFR, MET, FOXO3, NFKB2, SKI etc. (Fig. 3h). The downregulated genes were also highly enriched several terms reported to improve cancer cell growth, including angiogenesis, mitotic cytokinesis, positive regulation of cell proliferation, positive regulation of blood vessel endothelial cell migration (Fig. 3i). More importantly, approximately one-third of their regulated genes were overlapped, indicating that miR-574-5p shares with miR-574-3p a set of common target genes through direct or indirect pathways, which are critical for gastric carcinogenesis (Fig. 3j).

**Regulatory network of miR-574-5p/-3p targets in GC cells**. All the above findings indicated miR-574-5p/-3p played opposite functions in GC by suppressing different targets. The next question is who triggers the contrary changes of miR-574-5p/-3p arm in gastric carcinogenesis. Both target-mediated miRNA protection (TMMP) and target RNA directed miRNA

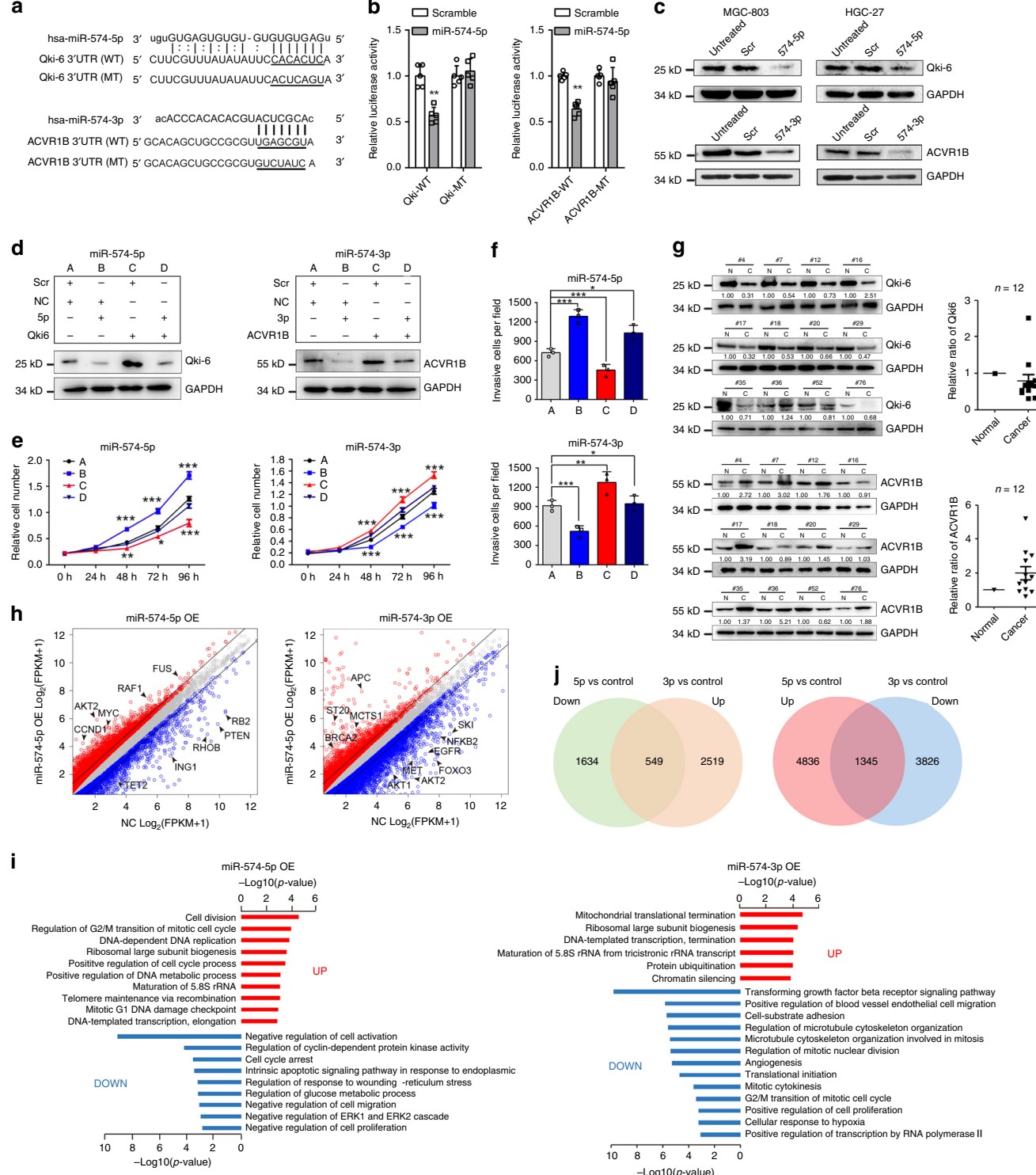

**Fig. 3** miR-574-5p/3p targeted endogenous QKI6 and ACVR1B, respectively. **a** Schematic representation of QKI6 and ACVR1B 3'UTRs showing putative miR-574-5p/-3p binding site. **b** Relative luciferase activity of wild type (WT) QKI6 or ACVR1B 3'UTRs constructs and miRNA binding site mutated constructs (MT) in 293T cells treated with scramble or miR-574-5p/-3p mimics. Three technical replicates from a single experiment representative of three independent experiments. **c** The protein level of QKI6 and ACVR1B in GC cells treated with scramble or miR-574-5p/-3p mimics. **d** The protein level of QKI6 and ACVR1B in MGC-803 cells co-transfected with microRNA mimics and QKI6 or ACVR1B overexpression constructs. **e** Cell proliferation analysis of MGC-803 cells in different groups shown in panel (**d**). Three technical replicates from a single experiment representative of two independent experiments. **f** The relative invasive cells of MGC-803 cells in different groups shown in panel (**d**). Three technical replicates from a single experiment representative of two independent experiments. **g** The protein level of QKI6 and ACVR1B in 12 GC tissues compared with the adjacent normal tissues. **h** Differential expressed genes in MGC-803 cells treated with miR-574-5p/-3p mimics. **i** GO and KEGG pathway analysis of differential expressed genes induced by miR-574-5p/-3p overexpression. **j** Comparison of differential expressed genes induced by miR-574-5p and miR-574-3p. Data are shown as means ± s.d. *$p < 0.05$, **$p < 0.01$, ***$p < 0.001$, Student's $t$-test.

degradation (TDMD) provided the possible theoretical basis for miRNA targets modulating miRNA arm-selection preference. To systematically uncover RNA targets bound by miR-574-5p/-3p in GC cells, we performed biotinylated miRNA pull-down. Biotinylated miR-574-5p/-3p or scramble mimics were transfected in MGC-803 cells, then lysed and incubated with streptavidin-coated beads to capture miRNA–RNA complexes. The RNA fraction of the captured complex was proceeded to RNA sequencing. The ratio of the abundance of pulled down RNAs compared with the input RNAs for cells transfected with Bio-miR-574-5p/-3p versus Bio-scramble mimic was calculated for enrichment ratio (ER) (Fig. 4a, Supplementary Data 2). We detected miR-574-5p/-3p mostly enriched protein-coding RNAs, and an additional fraction of RNAs, including lncRNAs, pseudogenes, snoRNA and others (Fig. 4b). By analyzing the differential expression of these enriched genes using the corresponding input RNAs, we found that the decline of the enriched targets for miR-574-5p/-3p in miR-574-5p/-3p overexpressed cells were more obvious when compared with all genes or miRanda-predicted miRNA targets. Moreover, a greater proportion of highly downregulated genes were present with the increased cutoff for ER, suggesting that genuine miRNA targets tended to exist in highly enriched mRNAs (Fig. 4c). In line with that, QKI6 and ACVR1B were also enriched for miR-574-5p and miR-574-3p pulled down RNAs, respectively. Several reported miR-574-5p/-3p targets in other kind of cancers were also suppressed and pulled down by miR-574-5p/-3p (Supplementary Fig. 3A). In addition, GO and KEGG analysis showed that miR-574-5p pulled down RNAs were highly enriched several terms reported to suppress cancer cell growth, including cell cycle arrest, positive regulation of apoptotic process and negative regulation of cell proliferation, negative regulation of cell migration, suggesting miR-574-5p played oncogenic role in GC through targeting this serious of genes (Fig. 4d, Supplementary Fig. 3B). On the contrary, miR-574-3p pulled down RNAs were highly enriched several terms possibly involved in cell growth and proliferation, including positive regulation of transcription, ribosome biogenesis, positive regulation of cellular metabolic process (Fig. 4d, Supplementary Fig. 3C). More notably, downregulated genes upon miR-574-5p or 3p introduction were situated in the central of the regulation network.

Among the highly enriched genes, we selected 12 genes (of which 6 were pulled down by miR-574-5p and 6 by miR-574-3p) for in vitro validation through qRT-PCR. For all of the genes tested, we confirmed an enrichment with Bio-miR-574-5p/-3p compared with the scramble control (Fig. 4e). Among these miR-574-5p/-3p targets, it was worth noting that two miR-574-5p targets: IBA57-AS1 and KLRC2, and two miR-574-3p targets: S100A1 and TMEM54, possessed sequence highly complementary to miR-574-5p or -3p, not only in the canonical seed sequence as that in QKI6 and ACVR1B (Supplementary Fig. 3D). To further certify the interaction of these four targets with miRNA, ribonucleoprotein immunoprecipitation (RNP-IP) based on Ago2 was employed. As expected, all of the target RNAs in Ago2-IP complex were increased upon miR-574-5p or 3p overexpression, indicating these targets were bound by Ago2 complex and the binding was increased with the introduction of miRNAs (Supplementary Fig. 3E).

**miR-574-5p/-3p highly complementary targets induce miRNA decay**. To test whether the highly complementary targets can trigger miRNA degradation through TDMD, we firstly constructed artificial miR-574-5p/-3p sponges expressing RNAs with six binding sites (6× target) that pair extensively with the seed and 3′-end of the miRNAs but contain a central bulge (Fig. 4f). As

expected, overexpression of artificial miR-574-5p sponges in GC cells decreased miR-574-5p level without affecting the primary-miR-574 expression. Similarly, miR-574-3p sponges decreased miR-574-3p level but not influenced the primary-miR-574 level (Fig. 4g, Supplementary Fig. 3F). Consistent with previous reports[14–17], these results showed that highly complementary target RNAs decreased miRNA level probably by triggering their degradation. Moreover, expression of miR-574-5p/-3p sponges increased the luciferase reporter activity of miR-574-5p/-3p's canonical targets, QKI6 and ACVR1B, respectively, in 293T cells (Fig. 4h), as well as increased their RNA and protein levels in two GC cell lines (Fig. 4i, Supplementary Fig. 3G).

Then we asked if the previously enriched endogenous highly complementary targets can decrease the corresponding miRNA expression in GC cells. To answer it, we focused on four highly enriched RNAs in Ago2-IP, IBA57-AS1, KLRC2, S100A1, and TMEM54, and overexpressed their potential miRNA binding sites containing fragments in MGC-803 cells (Fig. 5a, Supplementary Fig. 4A). As expected, overexpression of IBA57-AS1 and KLRC2 significantly decreased miR-574-5p level but not influence the primary-miR-574 expression level, respectively (Fig. 5b, Supplementary Fig. 4B). Similar results were gained for S100A1 or TMEM54 overexpression on miR-574-3p and primary-miR-574 (Fig. 5b, Supplementary Fig. 4B). In line with the changes of mature miRNA level, these highly complementary targets elevated the expression of QKI6 and ACVR1B (Fig. 5c).

To further evaluate whether the four highly complementary targets can induce the corresponding miRNA decay in GC cells, we conducted small RNA sequencing. The results showed that the total miR-574-5p reads decreased dramatically when IBA57-AS1 or KLRC2 was overexpressed, while proportion of the tailing and trimming miR-574-5p was increased apparently, especially the trimming ones, suggesting the miRNA decay induced by the two targets. In line with this finding, when S100A1 or TMEM54 was overexpressed, total miR-574-3p reads declined accompanied by slightly upregulation of the tailing and trimming miR-574-3p (Fig. 5d, Supplementary Data 3). The small RNA-sequencing data together with the unchanged expression of primary-miR-574 transcripts indicated that IBA57-AS1, KLRC2, S100A1, and TMEM54 indeed induced miR-574 decay. In addition, we also examined whether the modulation of IBA57-AS1, KLRC2, S100A1, and TMEM54 expression in GC cells can result in a global change of miR-574-5p/-3p potential targets. As a result, when IBA57-AS1, KLRC2 was overexpressed, the potential targets of miR-574-5p, defined as highly enriched in miR-574-5p pulled down RNAs and decreased with miR-574-5p over-expression (indicated in Fig. 4d), were upregulated when compared with all genes ($p = 0.012$, $p = 0.0099$) (Fig. 5e, Supplementary Data 4). When S100A1 and TMEM54 were silenced with siRNAs (Supplementary Fig. 4C, D), miR-574-3p increased statistically significantly but not impressively, indicating endogenous S100A1 or TMEM54 can induce miR-574-3p decay to some extent (Supplementary Fig. 4C). The potential targets of miR-574-3p, defined as highly enriched in miR-574-3p pulled down RNAs and decreased with miR-574-3p overexpression (indicated in Fig. 4d), were indeed repressed obviously when compared with all genes ($p = 0.0087$, $p = 0.0071$) (Fig. 5e, Supplementary Data 4). Meanwhile, the expression of partial of the most potential targets was also validated with quantitative PCR. The results indicated that 61% (25/41) and 66% (27/41) of the potential miR-574-5p targets were increased with IBA57-AS1, KLRC2 overexpression, respectively. Eighty-seven percent (33/38) and 79% (30/38) of the potential miR-574-3p targets were inhibited by knocking down S100A1 and TMEM54, respectively (Supplementary Fig. 4E, F). These results together demonstrated that the non-canonical highly complementary targets could

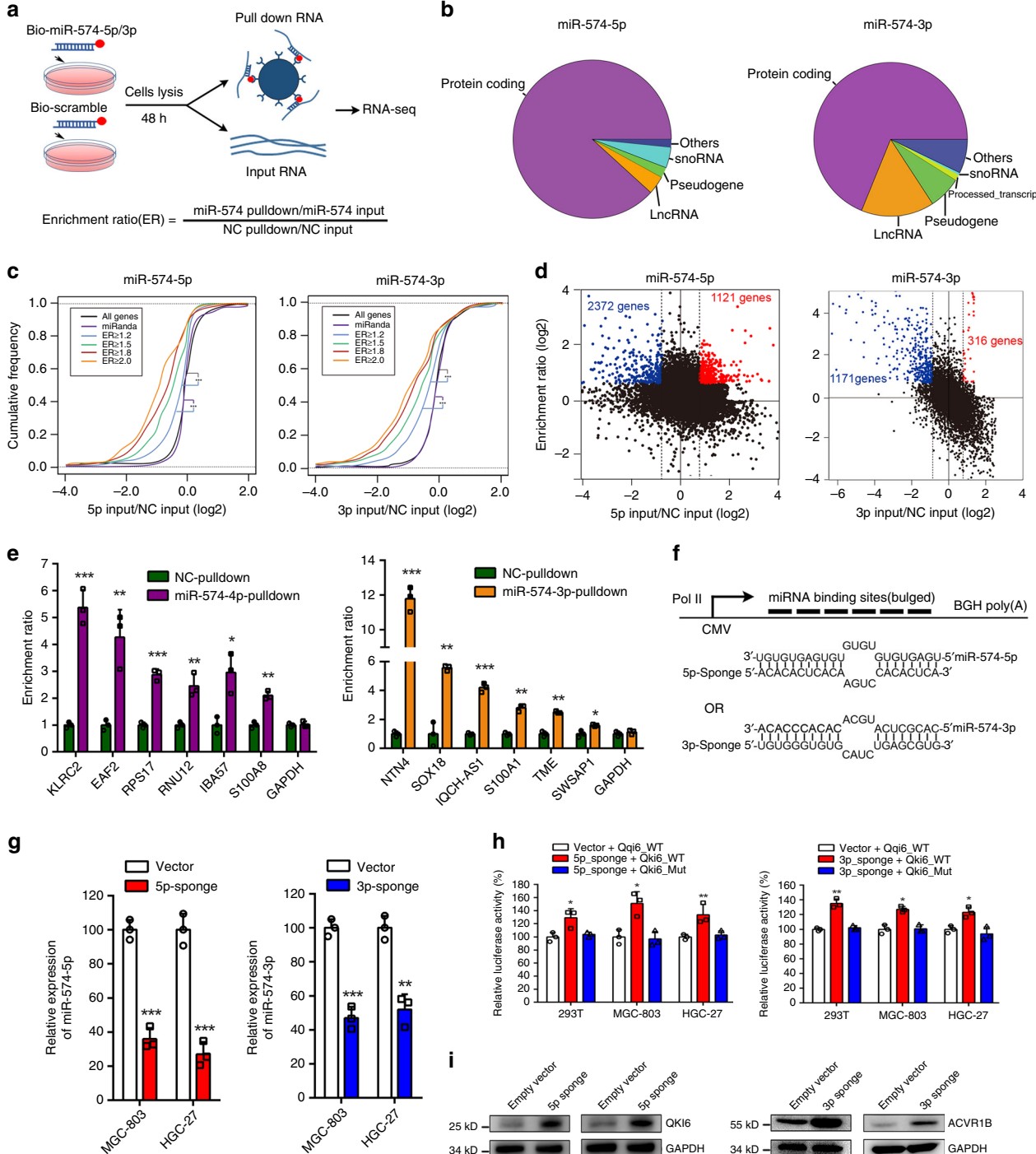

**Fig. 4** Systematic identification of miR-574-5p/-3p targets in GC cells. **a** Schematic representation of biotinylated miRNA pull-down experiment. **b** Classification of miR-574-5p/-3p enriched targets. **c**, **d** The expression change of miR-574-5p/-3p pulled down targets versus all genes or miR-574-5p/-3p targets predicted by miRanda in miR-574-5p/-3p overexpressed cells. ER: enrichment ratio. **e** qRT-PCR validation of the enrichment of the potential targets in miRNA pulled down experiment. **f** Schematic representation of artificial miR-574-5p/-3p sponges expressing RNAs with six binding sites (6× target) that pair extensively with the seed and 3′-end of the miRNAs but contain a central bulge. **g** Relative expression of miR-574-5p/-3p in GC cells with overexpression of artificial miR-574-5p/-3p sponges. **h** Relative luciferase activity of wild type QKI6 or ACVR1B 3′UTRs constructs in 293T and GC cells simultaneously treated with artificial miR-574-5p/-3p sponges. **i** The protein level of QKI6 or ACVR1B in GC cells with overexpression of artificial miR-574-5p/-3p sponges. Three technical replicates from a single experiment representative of two independent experiments. Data are shown as means ± s.d. **\*\*p < 0.01, \*\*\*p < 0.001, Student's t-test.

induce miRNA decay and thereby led to global change of their targets. It is also possible that target competition mediated by the highly complementary targets may contribute to the global change of their targets. In addition, we determined whether the two miR-574 arms were mutual regulated as they were quite complementary and found that miR-574-3p was not changed when miR-574-5p was overexpressed or suppressed, and vice versa (Supplementary Fig. 4G).

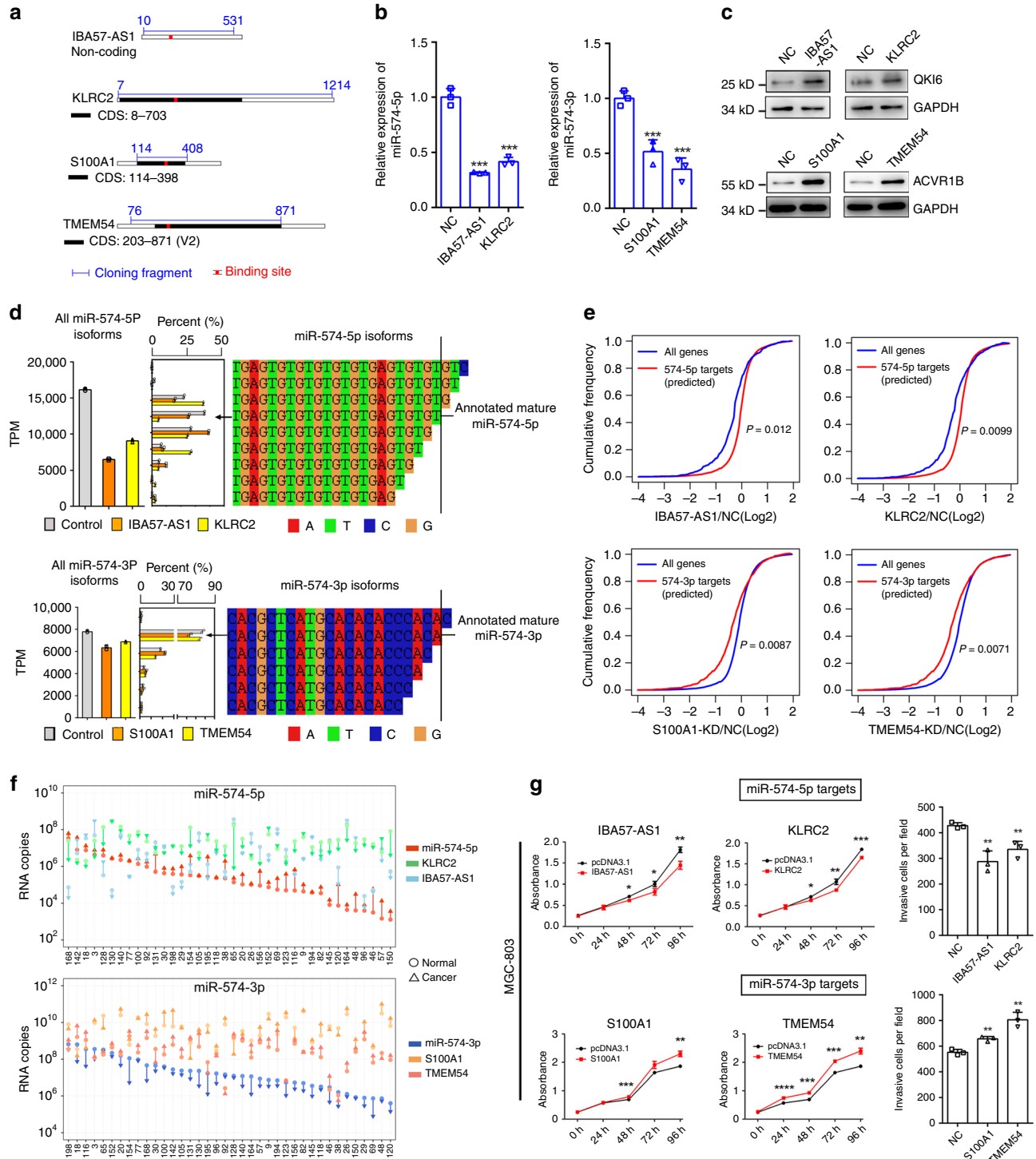

**Fig. 5** miR-574-5p/-3p highly complementary targets can induce miRNA decrease. **a** Schematic representation of the four targets, including the miR-574 binding sites and cloning fragment for overexpression. CDS, coding sequence. **b** Relative expression of miR-574-5p/-3p in GC cells with overexpression of IBA57-AS1, KLRC2 or S100A1, TMEM54. **c** The protein level of QKI6 or ACVR1B in GC cells with overexpression of IBA57-AS1, KLRC2 or S100A1, TMEM54. **d** Left panel: Bar plot indicating the absolute miRNA expression levels in MGC-803 cells with overexpression of the targets. Middle panel: the proportion of different miRNA isoforms. Right panel: isoform sequences displayed by color-coding. **e** Global expression of the potential miR-574 targets in MGC-803 cells with IBA57-AS1, KLRC2 overexpression and S100A1, TMEM54 knock down, respectively. **f** The absolute RNA copy number of the four targets (IBA57-AS1, KLRC2, S100A1, and TMEM54) and miR-574-5p/-3p in gastric cancer patients with previously identified inversely differential expression of miR-574-5p/-3p. **g** The proliferation rate (left) and invasion ability (right) of MGC-803 cells with overexpression of IBA57-AS1, KLRC2, S100A1, and TMEM54. Three technical replicates from a single experiment representative of two or three independent experiments. Data are shown as means ± s.d. *$p < 0.05$, **$p < 0.01$, ***$p < 0.001$, Student's $t$-test.

To evaluate the relevance of these highly complementary targets in GC development, we examined their absolute RNA copies in 46 pairs of GC tissues with previously identified inversely differential expression of miR-574-5p/3p (Fig. 1c). The results showed that the absolute copy number of these targets was massive compared with miRNAs, suggesting the expression levels of these targets could reach the level that were sufficient to induce miRNA decay. Meanwhile, we also observed that miR-574-5p targets (IBA57-AS1, KLRC2) was clearly downregulated, while miR-574-3p targets (S100A1 and TMEM54) was upregulated in these patients, which is in contrast to the corresponding miRNA change, suggesting the potential miRNA decay might be induced by these targets in gastric cancer patients (Fig. 5f). Functional analysis of these four targets by overexpressing them in MGC-803 and HGC-27 cells demonstrated that miR-574-5p targets IBA57-AS1 and KLRC2 obviously suppressed cell proliferation and invasive cell number while the two miR-574-3p targets S100A1 and TMEM54 did the opposite (Fig. 5g, Supplementary Fig. 4H-I). Taken together, these findings suggested the important roles of the highly complementary targets (such as IBA57-AS1, KLRC2, S100A1, and TMEM54) of miR-574-5p/-3p in gastric carcinogenesis. Since these targets can decrease miR-574-5p/-3p expression possibly by inducing miRNA decay and thus their distinct distribution in GC patients mediated the reverse expression changes of miR-574-5p/-3p, our results shed light on the importance of target-miRNA homeostasis.

**Arm-imbalance of miR-574 accelerates GC progression**. Target-miRNA homeostasis was maintained in physiological conditions, in which two arms of miR-574 precursor were considered to be balanced. Once target-miRNA homeostasis was broken in GC, the two arms of miR-574 shifted to show miR-574-5p upregulation and miR-574-3p downregulation, which we called arm-imbalance. As they had opposite roles in gastric carcinogenesis, whether the arm-imbalance of miR-574 accelerated GC progression? To test this hypothesis, we introduced miR-574-5p mimics and miR-574-3p inhibitors at multiple combinations in GC cells to mimic the arm-imbalance (Supplementary Fig. 5A, B). As expected, miR-574-5p overexpression (OE) combined with miR-574-3p inhibition (IN) promoted cell proliferation and invasion most obviously in both MGC-803 and HGC-27 cells. Instead, miR-574-5p-IN accompanied with miR-574-3p-OE repressed GC cell proliferation and invasion while promoted their apoptosis most significantly (Fig. 6a–d, Supplementary Fig. 5C–G, Supplementary Fig. 6A, B). Consistent with the in vitro observation, miR-574-5p-OE combined with miR-574-3p-IN accelerated the growth of MGC-803-engrafted tumors most significantly (Fig. 6e); the volumes and weights of tumors were maximal, while that of miR-574-5p-IN combined with miR-574-3p-OE group was minimal (Fig. 6f, g). Similarly, miR-574-5p-OE accompanied with miR-574-3p-IN promoted metastasis of gastric cancer cells to lung most remarkably, while miR-574-5p-IN combined with miR-574-3p-OE seriously impaired metastasis of the cells most obviously (Fig. 6h). The number of nodule formation in lungs of mice injected with miR-574-5p-OE + miR-574-3p-IN treated cells was significantly more than the others, whereas that of miR-574-5p-IN combined with miR-574-3p-OE was significantly less than the other groups (Fig. 6i, j, Supplementary Fig. 6C).

Next, we wanted to know if the arm-imbalance of miR-574 is responsible for GC progression in individual patients. We assigned each patient a risk score according to a linear combination of the expression level of miR-574-5p and miR-574-3p, that is, more significant miR-574-5p up and more significant miR-574-3p down in GC tissues reach a higher risk score, and showed the risk-score distribution of patients (Fig. 7a, b). We found that patients with higher risk score tended to have shorter survival time. Collectively, these findings strongly indicated arm-imbalance of miR-574 contributed to GC tumorigenesis and accelerated GC progression. We also performed a comprehensive expression analysis of two arms of a miRNA (miR-5p and miR-3p) in gastric cancer tissues using the entire TCGA gastric cancer database (424 patients) combined with another miRNA profiling dataset which included 73 gastric cancer patients and 94 normal controls (GSE61741). A total of 1749 miRNAs were included in this analysis and simultaneous detection of both arms was found in 838 miRNAs. Among them, 301 (35.91%) miRNA pairs exhibited same expression tendencies in gastric cancer compared with normal controls, while 76 (9.07%) miRNA pairs including miR-574 showed opposite expression alternation (Supplementary Fig. 7, Supplementary Data 5). Thus, we can tell that miR-574 is not the only miRNA with which a miRNA arm-selection was observed in gastric cancer and the target RNA directed miRNA degradation (TDMD) demonstrated here may be one of the underlying mechanisms.

Finally, we sought to further characterize the integral role of miR-574-5p/-3p and their enriched targets in GC tumorigenesis. We constructed a regulatory network of miR-574-5p/-3p and their enriched targets, in which, we classified the pulled down targets into highly complementary ones (inner circle) and the others (outer circle) according to the sequence complementarity to the microRNA, and also investigated their expression in gastric cancer in TCGA database or from our in-house qRT-PCR data (Fig. 7c). The majority of miR-574-5p enriched targets, including both the highly complementary targets and the others, decreased in GC tissues. This observation demonstrated that the decrease of multiple highly complementary targets led to upregulation of miR-574-5p partially through triggering miRNA decay, and in turn, miR-574-5p functioned through negatively regulating a series of canonical targets to decrease their expression in GC. On the contrary, the majority of miR-574-3p enriched targets were increased in GC tissues, which mediated miR-574-3p's down-regulation and also facilitated its tumor suppressor role in GC. In addition, we also noted that the canonical miRNA targets and these non-canonical targets were closely related for both miR-574-5p and -3p.

In summary, target-miRNA homeostasis was balanced in normal gastric cells, and once this homeostasis was broken by the key non-canonical highly complementary targets, two arms of miR-574 shifted. miR-574-5p rose to repress a series of tumor suppressive targets, while miR-574-3p declined to release multiple oncogenes to facilitate GC cell proliferation and invasion synergistically, finally accelerating GC development (Fig. 7d).

## Discussion

It is known that in some cases, both miRNA and miRNA* are functional and target different RNA populations[12,40,41]. Their functions could be similar or different depending on their targets. During miR-574 biogenesis, both miR-574-5p and miR-574-3p were accumulated in vivo but were reported to have opposite regulatory effects in various cancer developments[18–27]. However, how these two miRNAs from the same precursor played reverse functions in the same disease progression, and how the balance of their effects was maintained? Our finding uncovered that miR-574-5p and miR-574-3p indeed play contrary roles in GC progression. During gastric carcinogenesis, the oncogenic miR-574-5p rose meanwhile tumor suppressive miR-574-3p declined to synergistically facilitate and promote GC development although they had opposite roles. The transform of their arm preference in

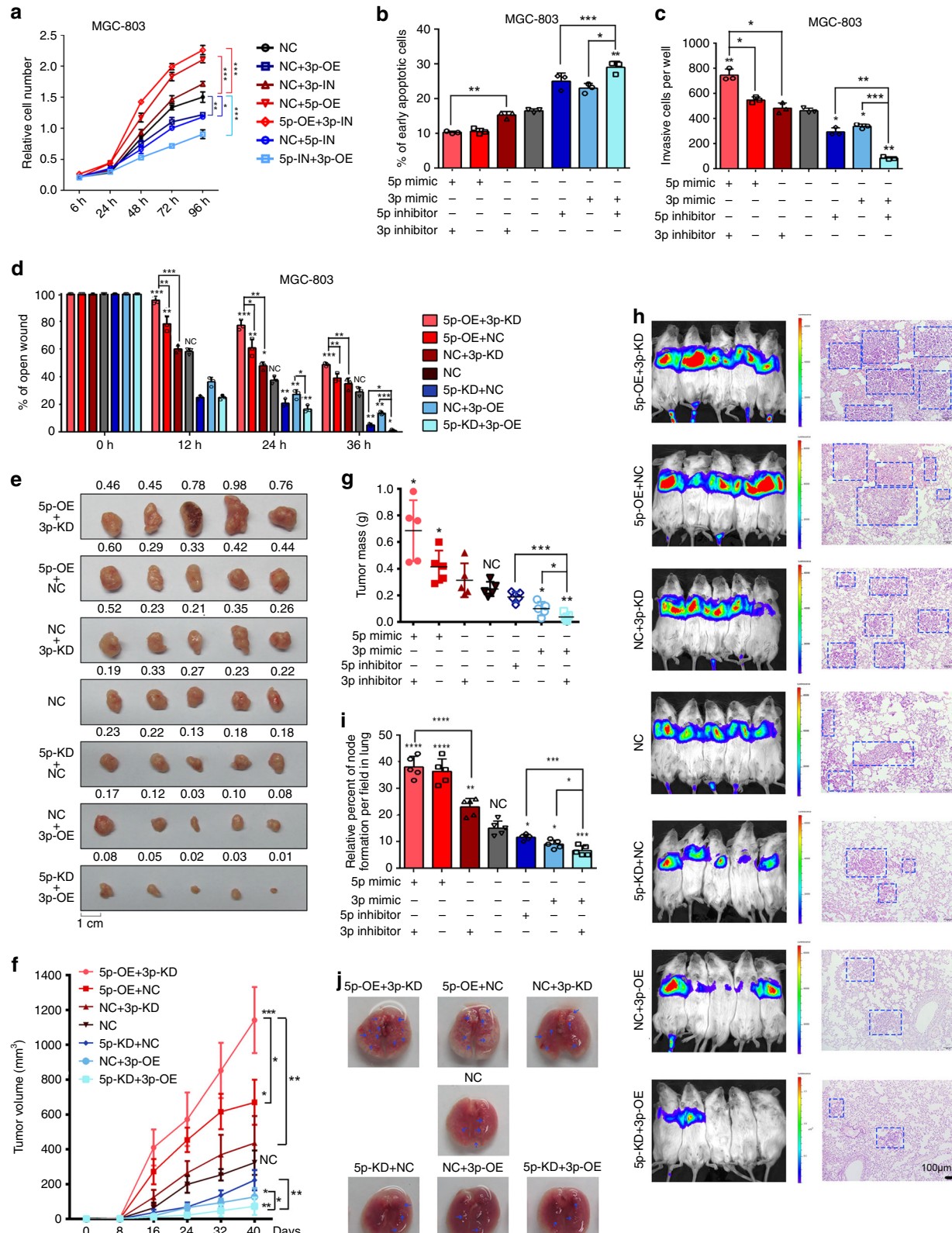

**Fig. 6** Arm-imbalance of miR-574 promoted GC progression. MGC-803 cells were treated with miR-574-5p/3p mimics (5p-OE/3p-OE) and miR-574-5p/3p inhibitors (5p-KD/3p-KD) at multiple combinations to mimic the arm-imbalance. The proliferation rate (**a**), percentage of apoptosis (**b**), invasive cell number (**c**), and wound closure percentage (**d**) of MGC-803 cells of the above seven groups was analyzed. Three technical replicates from a single experiment representative of two independent experiments. **e** Photographs of MGC-803-engrafted tumors in the seven different groups ($n = 5$). **f** Graph representing tumor volumes. **g** Tumor mass of the seven groups at the end of the experiment (day 40). **h** Bioluminescence imaging of mice showed GC cell metastasis in vivo and the typical H&E staining of the lung tissues. **i** Number of metastatic nodules formation in lungs. **j** Representative pictures of the lungs from mice in (**h**). Arrows indicated nodules. Data are shown as means ± s.d. *$p < 0.05$, **$p < 0.01$, ***$p < 0.001$, Student's $t$-test.

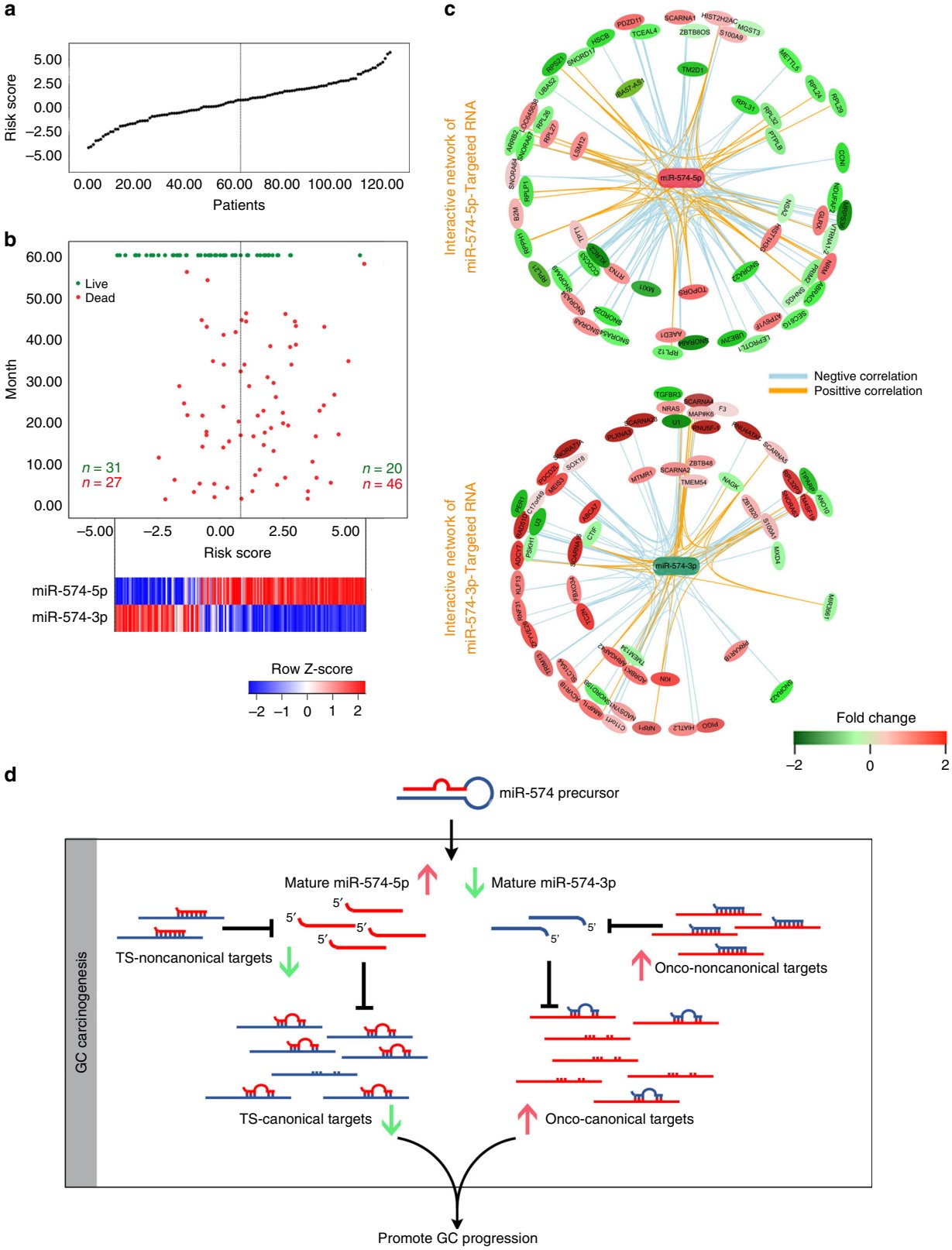

**Fig. 7** Arm-imbalance of miR-574 was correlated with poor prognosis of GC patients. **a** Risk-score distribution of the 124 GC patients. **b** Survival status and time of patients with according risk scores, and the miR-574-5p/3p expression profiles of patients with GC. **c** Network of miR-574-5p/-3p and their potential targets during gastric carcinogenesis. **d** Working model of arm-imbalance of miR-574 in promoting GC progression. Ts: tumor suppressor; Onco: oncogenic.

GC was found to be mediated partially by their targets, suggesting that target-miRNA homeostasis was accurately regulated. When the target-miRNA homeostasis or miRNA arm-balance was broken, the miRNA function changed and may lead to disease progression. It was noticeable that alternative miRNA arm preference increased biodiversity of a single miRNA gene, similar to alternative splicing of mRNAs, capable of responding to particular conditions quickly. Alternative miRNA arm preference may be a ubiquitous and common way of miRNA functioning, which needed to be verified further.

So far, mechanism of dynamic miRNA arm preference was less clear. Both the extent of complementarity and the abundance ratio of miRNA/target were crucial for the efficient decay of miRNA in the Target RNA-directed miRNA degradation (TDMD) theory[14–16]. Canonical miRNA binding sites, which lacked extended complementary to the 3′-end of miRNA, cannot induce TDMD efficiently. As reported, a two-nucleotide mismatch to the miRNA 3′-end would reduce the degree of TDMD dramatically, and a total of four mismatches completely abolished TDMD[15,16]. However, we found that a two-nucleotide mismatch to the miRNA 3′-end but highly complementary to the 3′-end adjacent region also can efficiently trigger miRNA decay. Whether the decay was dependent on the nucleotide addition and exonucleolytic degradation was not studied carefully here. In addition, it was likely there were a class of analogous targets contributing to the arm preference of miR-574 shift in GC tumorigenesis not just the several targets shown in our study. Because miRNA-targets homeostasis was maintained by a network in which miRNAs, multiple canonical miRNA targets and those non-canonical targets were mutual regulated and balanced exquisitely, not a point-to-point regulation.

miRNA-targets homeostasis was crucial for cellular metabolism and controllable proliferation[13,42]. When the key factors involved in this network changed, the homeostasis would be broken and cells would be out of control and cause disease. We found that miR-574-5p/-3p mediated miRNA-targets network played important roles in GC development. During gastric carcinogenesis, a series of tumor suppressive mRNAs targeted by miR-574-5p decreased and then led to upregulated miR-574-5p, which further repressed many tumor suppressive targets, forming a positive feedback to promote GC progression and even higher levels of miR-574-5p. Meanwhile, miR-574-3p functioned conversely and decreased significantly in GC through the opposite feedback way. Along with the GC progression, the balance slanted more severely and cells were out of control and more aggressive. Therefore, targets mediated shift of miR-574 arm-preference may not be the driver but a crucial mediator of gastric carcinogenesis and progression. Due to the strong correlation of miR-574-5p/-3p expression pattern with the survival rate of GC patients, miR-574-5p/-3p ratio may be developed as prognostic indicator to predict the therapeutic responses of GC. In view of the extremely remarkable effects of miR-574 in regulating GC progression in vitro and in vivo, along with the great progress in RNA-derived therapeutics[40], miR-574-5p inhibitors combined with miR-574-3p mimics would be developed as promising therapy drug to treat GC in the future.

In conclusion, our findings indicated that miR-574 arm-imbalance mediated partially by its targets contributed to GC progression. Re-modification of the miR-574-targets homeostasis may represent a realistic approach for gastric cancer prevention and therapy.

## Methods

**Clinical specimens and cell lines**. Primary gastric cancer tissues and their adjacent non-tumorous gastric samples from 167 patients were collected from Cancer Institute and Hospital, Chinese Academy of Medical Sciences and Shanxi Cancer Hospital from 2009 to 2017. Fresh samples were snap frozen in liquid nitrogen immediately after resection and stored at −80 °C. The study was approved by the ethical board of Cancer Institute and Hospital and the ethical board of the Institute of Basic Medical Sciences, Chinese Academy of Medical Sciences and all samples were obtained with patients' informed content and were histologically confirmed by staining with hematoxylin–eosin. Sixty-six cases of formalin-fixed paraffin-embedded (FFPE) GC tissues FFPE samples were collected from Cancer Institute and Hospital, Chinese Academy of Medical Sciences and conducted follow-up studies. The patient characteristics were provided in Supplementary Tables 1 and 2. 293T and gastric cancer cell line, MGC-803 and HGC-27 were obtained from the ATCC and grown in DMEM with 10% FBS (Hyclone, Logan, Utah) at 37 °C in 5% $CO_2$ cell culture incubator. No cell lines used in this study were found in the database of commonly misidentified cell lines that is maintained by ICLAC and NCBI Biosample. Cell lines were tested for mycoplasma detection according to the ATCC cell line verification test recommendations.

**Quantification of RNA and protein**. Total RNA was extracted from cells and tissues using TRIzol (Invitrogen, Carlsbad, CA, USA) according to the manufacturer's instruction. The RNA was quantified by absorbance at 260 nm. To assess the levels of miR-574s or its targets, quantitative-PCR analysis was conducted in the Bio-Rad IQ5 Q-PCR system (Bio-Rad, Hercules, CA) according to the manufacturer's instruction. The data were normalized using endogenous U6 snRNA, GAPDH or β-actin. The expression levels of miRNAs in cancer relative to its non-tumorous control and GC cells were calculated using the equation $2^{-\Delta\Delta CT}$ in which $\Delta C_T = C_T\ 574s\text{-}C_T\ U6$. The value of the relative expression ratio < 1.0 was considered as low expression in cancer relative to the non-tumorous control, where others were considered as high expression. Primer sequences are listed in Supplementary Table 4.

For western blotting, cells were washed twice with ice-cold phosphate-buffered saline (PBS) and ruptured with RIPA buffer (Pierce, Rockford, IL) containing 5 mM EDTA, PMSF, cocktail inhibitor, and phosphatase inhibitor cocktail. Cell lysates (20 μl) were resolved by SDS-PAGE and transferred onto PVDF membranes. Membranes were blocked for 1 h with 5% skim milk in Tris-buffered saline containing 0.1% Tween 20 and incubated overnight at 4 °C with antibodies included those against QKI6 (1:1000 dilution) (AB9906, Merck Millipore, Germany), ACVR1B (1:1000 dilution) (ab109300, Abcam, Cambridge, MA), and GAPDH (1:5000 dilution) (60004-1-Ig, Proteintech, Rosemont, IL). Membranes were washed 30 min with Tris-buffered saline containing 0.1% Tween-20, incubated for 1 h with appropriate secondary antibodies conjugated to horseradish peroxidase, and developed using chemiluminescent substrates. The uncropped and unprocessed scans of the blots were provided in Supplementary Fig. 8.

**Determination of gene copy numbers**. The qRT-PCR product of miR-574-5p and miR-574-3p, and pMIR-REPORT-KLRC2, pMIR-REPORT-IBA57-AS1, pMIR-REPORT-S100A1, and pMIR-REPORT-TMEM54 plasmids were used as standard. A 10-fold serial dilution series of the PCR product and plasmids, ranging from $1 \times 10^1$ to $1 \times 10^{10}$ copies μg$^{-1}$ of total RNA, was used to construct the standard curves. The concentration of the PCR products and plasmids were measured using a Nanodrop One (Thermo Fisher) and the corresponding copy number was calculated using the following equation: DNA (copy) = $6.02 \times 10^{23}$ (copy mol$^{-1}$) × DNA amount (g)/DNA length (bp) × 308.95 (g mol$^{-1}$) × 2. Ct values in each dilution were measured in triplicate using a qRT-PCR to generate the standard curves for miRNAs and targets, respectively. The Ct values were plotted against the logarithm of their initial template copy numbers. Absolute quantification was used to determine the exact copy concentration of target genes by relating the Ct value to a standard curve. The Ct values of miRNAs, targets and references (U6 or GAPDH) of all samples are obtained and the copy number of miRNAs and targets (copies μg$^{-1}$) was presented relative to the expression of reference genes.

**Oligonucleotides, constructs, and transfections**. The mimics of miR-574-5p, 574-3p, inhibitors, and negative controls were purchased from Ribobio (Guangzhou, China) and transfected into HGC-27 and MGC-803 using DharmaFECT 2 (Chicago, IL, USA) at a final concentration of 50 nM. The siRNA targeting S100A1 and TMEM54 (si_S100A1 and si_TMEM43) and control were purchased from Ribobio (Guangzhou, China) and transfected into HGC-27 and MGC-803 using Lipofectamine RNAiMAX Reagent (Thermo Fisher, San Jose, CA) according to the manufacturer's instructions. The open reading frames (ORFs), but not including the 3′ UTRs of QKI6 and ACVR1B, and the sequences containing miR-574 binding sites of KLRC2, IBA57-AS1, S100A1, and TMEM54 were obtained by PCR amplification using human complementary DNA (cDNA) as the template and inserted into pcDNA3.1(+). The wild-type and mutant 3′UTR of the QKI6 and ACVR1B mRNA, as well as the sequences containing miR-574 binding sites of KLRC2, IBA57-AS1, S100A1, and TMEM54 was cloned into pMIR-REPORT (Thermo Fisher). The PCR primers were described in Supplementary Table 4. The recombination expression plasmids were transfected into HGC-27 and MGC-803 cells using Lipofectamine LTX (Thermo Fisher) according to the protocol. The efficiency of transfection was confirmed by qRT-PCR and western blot.

**Cell proliferation, migration, and invasion assays**. To measure the effects on cellular proliferation rates, GC cells were incubated in 10% CCK-8 (DOJINDO) diluted in normal culture media at 37 °C until visual color conversion appears. Proliferation rates were determined at 12, 24, 36, 48, 60, 72 h post transfection, and quantification was done on a microtiter plate reader (Spectra Rainbow, Tecan) under manufacturer-recommended protocol.

For cell migration assay, GC cells were grown to confluence on 12-well plastic dishes and treated with miRNA mimics or siRNAs. Then 36 h post treatment, linear scratch wounds (in triplicate) were created on the confluent monolayers using a 200 µl pipette tip. Immediately after wounding (time 0) and at 12 h intervals for 36 h, images of the miRNA mimics, siRNAs-treated GC cells were taken using a Zeiss AxioCam MR digital camera mounted on a Zeiss Axiovert 40 CFL inverted light microscope (Carl Zeiss, Oberkochen, Germany). The percentage decrease in the wound gaps were calculated using Axiovision computer-assisted image analysis and normalized to the time 0 wounds. To remove cells from the cell cycle prior to wounding, mitomycin C (5 µg ml$^{-1}$) was added into the medium of miRNA mimics-treated GC cells after 2 h incubation. Linear scratch wounds were created and evaluated as described above.

For cell invasion assay, HGC-27 and MGC-803 cells were grown to confluence, and transfected with miR-574s or control mimics. Twenty-four hours post transfection, cells were seeded onto a Matrigel-coated membrane matrix (BD Bioscience) present in the insert of a 24-well culture plate. Fetal bovine serum was added to the lower chamber as a chemoattractant. After 24 h, the non-invading cells were gently removed with a cotton swab. Invasive cells located on the lower surface of chamber were stained with the 0.1% crystal violet (Sigma) and counted.

**Immunohistochemistry and light microscopy**. The gastric tissues were preserved in formalin solution, embedded in paraffin and sectioned. Sections were stained with hematoxylin and eosin (H&E). For immunohistochemistry, Endogenous peroxidase was blocked by incubation for 15 min in 0.3% H$_2$O$_2$. Antigen retrieval was carried out in 0.01 M sodium citrate-hydrochloric acid buffer solution. Anti-QKI6 (1:50 dilution) (AB9906, Merck Millipore), anti-ACVR1B (1:200 dilution) (ab109300, Abcam), anti-Ki-67 (1:100 dilution) (9449T, CST), and anti-caspase-3 (1:100 dilution) (9664T, CST) antibodies were used for immunohistochemical analysis. Tissues were incubated with the primary antibody at 4 °C overnight. After washing with phosphate-buffered saline (PBS), the cells were incubated with the appropriate secondary antibody for 30 min at 37 °C. Peroxidase activity was revealed by 3,3-diaminobenzidine and cells were counter-stained with hematoxylin. As a negative control, antibodies against rabbit IgG were used. Cells were viewed and photographed with a Zeiss UV LSM 510 confocal microscope.

**Tumorigenicity and metastasis formation assay**. All animal experiments were performed with the approval of the Research Ethics Committee of the Institute of Basic Medical Sciences, Chinese Academy of Medical Sciences. Tumorigenicity and metastasis formation assay were performed. In brief, 4–6-week-old female BALB/c and NOD/SCID mice were randomly divided into three or seven groups with four or five mice per group. Mimics of MiR-574-5p, 574-3p, inhibitors and negative controls transfected MGC-803 cells ($3 × 10^6$) were suspended in 100 µl PBS and then injected subcutaneously into either side of the posterior flank of the same female BALB/c athymic nude mouse at 5–6 weeks of age. Five nude mice were included in each group and tumor growth was examined every three days over a course of 4 weeks. Tumor volume ($V$) was monitored by measuring the length ($L$) and width ($W$) of the tumor with calipers and was calculated with the formula $V = (L × W^2) × 0.5$.

We used the NOD/SCID mice xenograft model to estimate the metastatic ability of the parental and infected MGC-803-luci cells (Genechem Co., Ltd, Shanghai, China) in vivo. Mimics of MiR-574-5p, 574-3p, inhibitors, and negative controls were transfected into MGC-803-luci cells. $1 × 10^5$ functional viable cells were resuspended in 0.1 ml phosphate-buffered saline and injected into the lateral tail vein. Five weeks after injection, we monitored tumor metastases using a live animal bioluminescence imaging system (PerkinElmer). To analyze the biodistribution of D-luciferin in vivo, the compounds were injected (100 µl of 5 mM solutions in PBS) into luciferase-expressing mice. Images were analyzed using Living Image software. Regions of interest (ROIs) were drawn around each cell mass, and the total numbers of photons within each ROI were recorded. ROI size was held constant across all images. Then mice were killed and the lungs and livers were extracted and fixed in 4% paraformaldehyde in phosphate-buffered saline. Paraffin embedding, sectioning and staining with hematoxylin–eosin were performed. Visible lung and liver metastases were measured and counted using a microscope.

**Dual luciferase reporter experiments**. The 3′UTR of the human QKI6 and ACRV1B mRNA was cloned in between the SpeI and HindIII sites of pMIR-REPORT (Thermo Fisher). Mutation of the QKI6 and ACRV1B 3′ UTR sequence was created using a QuickChange Site-Directed Mutagenesis kit (Stratagene). The miRNA sponge for miR-574-5p and 574-3p binding sites with 4-nt spacers for bulged sites was constructed into pcDNA3.1(+). Sequences of the primers were shown in Supplementary Table 4. miRNA mimics of miR-574-5p, 574-3p, and control miRNA mimic were obtained from Ribobio.

293T cells were seeded onto 24-well plates ($1 × 10^5$ cells per well) the day before transfections were performed. Cells (≈70% confluent) were transfected with pMIR-REPORT luciferase reports (50 ng) per well, pRL-TK control luciferase (10 ng) per well, and miR-574-5p and 574-3p mimic (50 nM). All transfections were carried out in triplicate with Lipofectamine LTX (Thermo Fisher). Cell lysates were prepared with Passive Lysis Buffer (Promega, Madison, WI) 48 h after transfection, and Firefly and Renilla luciferase activity were measured using Dual Luciferase Assay (Promega) according to the manufacturer's instructions.

**MicroRNA pull-down**. Biotinylated miRNA mimics were synthesized by Ribobio (Guangzhou, China), and transfected into MGC-803 cells. The cells were harvested 48 h post transfection and subjected to miRNA pull-down analysis. In brief, cells were rinsed twice with ice-cold PBS before harvesting in 10 mL ice-cold PBS by scraping. Then the cell pellets were resuspended in 500 µL Lysis buffer (20 mM Tris-HCl, pH 7.0, 100 mM KCl, 5 mM MgCl$_2$, 25 mM EDTA, 0.5% NP-40, 1 mM PMSF, proteinase inhibitor cocktail), and the supernatants were incubated with Dynabeads Streptavidin M-280 (Thermo Fisher). The beads were washed five times with lysis buffer with gentle rock and added with 100 µL lysis buffer. Total RNA was extracted with TRIzol reagent, and the purified RNA was tested for quality by Agilent 2100 Bioanalyzer (Agilent, Santa Clara, CA). Ribosome RNA depleted RNA libraries and sequencing were performed by Genergy Bio (Genergy Bio-Technology Co., Ltd., Shanghai, China) using an Illumina HiSeq3000 (Illumina, USA).

**Ago2 RIP**. MGC-803 cells were transfected with miR-574-5p, miR-574-3p or mimic control. After 48 h, cells were used to perform RIP experiments using an anti-Ago2 antibody (ab32381, Abcam) or a rabbit isotype IgG (12-370, Merck Millipore). RNAs were isolated from the immunoprecipitation products and quantified. Real-time PCR was performed to examine the expression levels of miR-574 targets.

**Gene dys-regulation after miR-574-5p (or 3p) overexpression**. Postalignment gene counts were generated using GFOLD (V1.1.4), a useful tool for read quantification. Differentially expressed genes were identified with DEGseq. The top hits with |log2FC| > log2(2) and FDR < 0.001 were flagged as candidates for downstream biological function analysis by DAVID (https://david.ncifcrf.gov/). The list of RNAs whose expression was changed upon miR-574-5p or miR-574-3p over-expression was shown in Supplementary table 5.

**Analysis of RNA-sequencing data**. The quality of raw sequencing reads were determined using FastQC (http://www.bioinformatics.babraham.ac.uk/projects/fastqc/) and adapters were trimmed by Trimmomatic (V0.33). Trimmed reads smaller than 50 nt were discarded and the rest were mapped to human reference genome (hg38) augmented with RefSeq genome annotation using Tophat (V2.1.0) with default parameters. The expression level of genes were quantified using Cufflinks in FPKM format. For RNA-seq following miRNA overexpression, the cumulative frequency of enrichment ratio {5p input/NC input} for all genes and 3p and 5p target genes were calculated, respectively. Genes with no FPKM > 0.1 in any samples were considered not expressed and were discarded in this analysis.

For miRNA pull-down sequencing, the enrichment ratio {Bi-miR-574-5p (or 3p) pull-down/Bi-cel-miR-67 pull-down}/{Bi-miR-574-5p (or 3p) input/Bi-cel-miR-67 input} was calculated for each gene. Genes with no FPKM > 0.1 in any one samples were considered not expressed and were discarded in this analysis. For informatic analysis of the pull-down data, genes whose enrichment ratio above 1.2 were considered as candidates. To test the expression levels of putative candidates, gene lists with different enrichment ratio cutoff, miRanda-predicted targets (directly downloaded from miRanda website) and all the expressed ones in miR-574-5p (or 3p) overexpression were plotted in a cumulative distribution function (CDF) plot, and the Kolmogorov–Smirnov test was used for statistical comparisons between them. The list of target RNAs pulled down by miR-574-5p/3p was shown in Supplementary Data 2.

**Analysis of the small RNA-sequencing data**. Reads with low complexity were filtered out based on their dinucleotide entropy (removing < 1% of the reads). Only reads with a minimum length of 14 nt were retained. Alignments to the miRNA database miRBase release 18 (http://www.mirbase.org/) were performed by the software bowtie (version 0.9.9.1) with parameters -v 2 -a -m 100, tracking up to 100 best alignment positions per query and allowing at most two mismatches. Reads that mapped to a miRNA but at the same time also mapped with fewer mismatches to the genome (ce6) were filtered out. The expression of each miRNA was determined by counting the number of associated reads. To compensate for differences in the read depths of the individual libraries, each sample was divided by its total number of counts and multiplied by the average sample size. The resulting values were log2 transformed using a pseudo-count of 1 ($y = log2(x + 1)$).

**Analysis of miRNA arms expression in gastric cancer**. To identify the different expression level of miRNA-3p and miRNA-5p, we analyzed stomach profiles between cancer and normal cases. We used two databases: The Cancer Genome

Atlas (TCGA) with 50 normal controls and 374 stomach patients and GSE61741 with 94 normal controls and 70 stomach patients. 1757 miRNAs were calculated and 838 of them has both 3p and 5p. Among them, 377 DEGs between normal controls and stomach patients and 76 of them expressed in different trend in 3p and 5p pairs.

**Statistical analysis**. For all studies, *n* per group is as indicated in the figure legend. Statistical differences in tissue miRNA expression levels between cancer and normal sample sources were determined using two-sided Mann–Whitney *U* test. Student's *t* test (two-tailed) was performed for two-group data and three-group data were analyzed using one-way analysis of variance. The Spearman's rank correlation test was conducted for statistical correlations. All data were analyzed using GraphPad Prism 5.0 software (GraphPad Software, Inc., USA) or SPSS 16.0 software (SPSS Inc., Chicago, IL, USA) and presented as means ± SD. No statistical methods were used to predetermine sample size. All experiments were carried out with at least three biological replicates. We chose the appropriate tests according to the data distributions. The experiments were not randomized and we did not exclude any samples. The investigators were not blinded to allocation during experiments and outcome assessment.

**Reporting summary**. Further information on research design is available in the Nature Research Reporting Summary linked to this article.

## Data availability

The RNA-sequencing data were available from the GEO database under accession number GSE112859, GSE133628, and GSE133629. All other data supporting the findings of this study are available from the corresponding author on request.

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

## Acknowledgements

This work was supported by the National Key Research and Development Program of China (2016YFA0100601 to J.Y., 2016YFA0100103 to Y.M.), CAMS Innovation Fund for Medical Sciences (2019-I2M-2-001 to J.Y., 2017-I2M-3-009 to X.W. and J.Y., 2018-I2M-1-001 to Z.X., 2016-I2M-3-002 to Y.M. and F.W., 2017-I2M-1-015 to H.Z.), the National Natural Science Foundation of China (81530007 and 31725013 to J.Y.; 31571523 and 31771444 to Y.M.), the Fundamental Research Funds for the Central Universities (3332019001), the Beijing Nova Program (2016B492 to Y.M.), the CAMS (2016GH310001 and 2017-I2M-B&R-04 to J.Y.; 2018RC310015 to Y.M.), Grant from

Medical Epigenetics Research Center, CAMS (2017PT31035) and the Chongqing Science and Technology Bureau of China (cstc2016shmszx1209; cstc2018jxjl130021).

## Author contributions

J.Y. and Y.M. supervised the study; Z.Z., J.P., and D.Z. designed and performed the experiments with help from X.W., S.Y., F.L., X.Z., H.Z., and F.W.; X.W. performed the clinical association analysis; J.X. and T.Z. performed the bioinformatics analysis with help from D.W., Y.M, J.Y., and X.W. wrote the paper.

## Competing interests

The authors declare no competing interests.
