## [Peer Review File · Nature Communications]

Reviewers' comments:

Reviewer #1, Expertise: miRNA, gene regulation (Remarks to the Author):

The paper entitled 'Targets mediated microRNA arm-imbalance promotes gastric cancer progression' investigates two miRNA arms from the same precursor, miR-574-5p and miR-574-3p, for their roles in gastric cancer progression. They found that a higher 574-5p plus lower 574-3p expression pattern was strongly correlated with higher TNM stages and shorter survival of patients. The arm-imbalance of miR-574 correlated with complementary targets in gastric carcinogenesis. Furthermore, the authors claim that non-canonical highly complementary targets regulate the expression of the two arms of miR-574, with reduced miR-574-3p levels resulting in derepression of multiple oncogenes to facilitate GC cell proliferation and invasion, and with increasing expression of miR-574-5p, which represses a series of tumor suppressive targets.

The manuscript is well written, data are clearly presented and the methodology is in general sufficiently described.

Major comments:

It is unclear why the authors focused on miR-574. Was this the only miRNA where a miRNA arm-selection was observed? Which miRNAs were screened? How many miRNAs showed an arm ratio variation? How strong was the variation for other miRNAs?

The associations of miR-574-5p and -3p in GC development as well as the functional characterization and target gene analysis are solid, however, the claim that highly complementary targets of miR-574-5p/-3p can induce miRNA decay and mediate arm-imbalance of miR-574-5p/-3p in GC progression relies entirely on associations and is not supported by sufficient data. The fact that overexpression of artificial sponges can lead to degradation of mature miRNAs is well known. What the authors need to convincingly show is that the expression levels of targets/ the number of miRNA binding sites in IBA57-AS1, KLRC2, S100A1 and TMEM54) reach levels in GC that are sufficient to induce miRNA decay. This cannot be demonstrated by relative expression studies.

Does silencing of IBA57-AS1, KLRC2, S100A1 and TMEM54 in GC cells result in a global repression of 574-5p targets?

To infer a mechanism of miRNA decay simply because miR-574-5p level were decreased without a concomitant decrease in primary-miR-574 transcripts is flawed, since other mechanisms may account for this. Did the authors notice tailing and trimming in their sequencing data or in Northern blots?

Hyperbels in the text should be avoided throughout the paper

Reviewer #2, Expertise: gastric cancer, miRNA
(Remarks to the Author):

This interesting manuscript demonstrates the role miR-574-5p and miR-574-3p in gastric cancer. Manuscript is quite extensive. The authors provided a number of lines of evidence demonstrating the opposite functions of miR-574-5p and miR-574-3p in gastric cancer as well as the critical function of the ratio of these two molecules. I believe that is the first or one of the first papers demonstrating that the ratio of two microRNAs derived from the same precursor is important in carcinogenesis.

Critique.

1. It is not clear why the authors started with isoforms of miR-219, miR-361 and miR-574. Did they do any preliminary screening?
2. First the authors identified 2 targets QKI6 and ACVR1B, and show their effect in gastric cancer. Then they did systematic identification a number of additional targets. If, QKI6 and ACVR1B account for all the effects of miR-574-5p and miR-574-3p as manuscript suggests, what is the reason then to speculate about roles of additional targets?
3. How many targets of miR-574-5p and miR-574-3p were higher on Targetscan list than ones they studied, and why the authors focused on QKI6 and ACVR1B?
4. There are over 100 publications about miR-574-5p and miR-574-3p, many of them about their role in cancer, and show that multiple other important targets of these two microRNAs are important in cancer (authors cited some of these papers). Are these targets expressed in gastric cancer, particularly vs QKI6 and ACVR1B? Are they inhibited by these two microRNAs? Were they identified by the authors in their systematic approach?
5. Previous studies identified QKI6 as a target of miR-574-5p (as mentioned by the authors). So these experiments including QKI6 are rather a confirmation of previous data in different cancer.
6. Introduction and discussion sections need to be shortened.

Reviewers' comments:

Reviewer #1, Expertise: miRNA, gene regulation (Remarks to the Author):

The paper entitled 'Targets mediated microRNA arm-imbalance promotes gastric cancer progression' investigates two miRNA arms from the same precursor, miR-574-5p and miR-574-3p, for their roles in gastric cancer progression. They found that a higher 574-5p plus lower 574-3p expression pattern was strongly correlated with higher TNM stages and shorter survival of patients. The arm-imbalance of miR-574 correlated with complementary targets in gastric carcinogenesis. Furthermore, the authors claim that non-canonical highly complementary targets regulate the expression of the two arms of miR-574, with reduced miR-574-3p levels resulting in derepression of multiple oncogenes to facilitate GC cell proliferation and invasion, and with increasing expression of miR-574-5p, which represses a series of tumor suppressive targets.

The manuscript is well written, data are clearly presented and the methodology is in general sufficiently described.

Major comments:

It is unclear why the authors focused on miR-574. Was this the only miRNA where a miRNA arm-selection was observed? Which miRNAs were screened? How many miRNAs showed an arm ratio variation? How strong was the variation for other miRNAs?

Reply: We appreciate this reasonable concern from the reviewer. Actually, the current study of the role of miR-574 in gastric cancer was initiated from another project which investigated promoter methylation level change in gastric cancer based on our previous MeDIP (Methylated DNA immunoprecipitation) combined with a human miRNA tiling microarray analysis (Rebuttal Figure 1A). These results were not

included in the current manuscript as it will be reported in another work, which mainly focused on the role of miR-369 in gastric cancer (Molecular Oncology, MOLONC-18-0804, under revision).

Briefly, we identified several potential miRNAs correlated with hypermethylation and up-regulated upon 5-Azacytidine (AZA) treatment in gastric cancer, which were miR-10a/b, miR-33b, miR-219, miR-369, miR-574 and so on (Rebuttal Figure 1B, C). Furthermore, the divergent role of miR-574-5p and miR-574-3p in various cancers was reported (Ref 1-10) as described in the Introduction section, which drove us curiosity to uncover the detail function of two arms of miR-574 in gastric cancer. We then examined their expression in many gastric cancer tissues compared with the adjacent non-cancerous tissues, as well as the expression of the two arms of miR-219 and miR-369.

Rebuttal figure 1 (A) Schematic procedure of DNA methylation-induced miRNA identification. (B) The correlation between specific miRNAs and hypermethylated regions. Red ellipses indicated the hypermethylated regions. Green ellipses indicated

the hypomethylated regions. Gray ellipses indicated no significant difference. The red arrow indicated miR-574. (C) AZA induced the expression of 13 miRNAs in a concentration-dependent manner in gastric cancer cells as shown by qRT-PCR analysis. The red arrow indicated miR-574-3p.

To further solve the reviewer's concern, we performed a comprehensive expression analysis of two arms of a miRNA (miR-5p and miR-3p) in gastric cancer tissues using the entire TCGA gastric cancer database (424 patients) combined with another miRNA profiling dataset which included 73 gastric cancer patients and 94 normal controls (GSE61741). A total of 1749 miRNAs were included in this analysis and simultaneous detection of both arms was found in 838 miRNAs. Among them, 301 (35.91%) miRNA pairs exhibited same expression tendencies in gastric cancer compared with normal controls, while 76 (9.07%) miRNA pairs including miR-574 showed opposite expression alternation (Rebuttal figure 2A). Thus, we can tell that miR-574 is not the only miRNA with which a miRNA arm-selection was observed, and 76 miRNAs showed an arm ratio variation in gastric cancer. Please refer to the heatmap presented in Rebuttal figure 2B and supplementary table 9. Anyway, the underlying mechanism of the dynamic miRNA arm-preference in gastric cancer remains unclear, but we believe the target RNA directed miRNA degradation (TDMD) demonstrated in the current manuscript might be one of them.

Finally, we rephrased the "Result" for selecting miR-574 as our research interest and the supplemented miRNA global analysis have been added in the revised manuscript.

Please refer to the revised Fig S7 and supplementary table 9 for related information.

Fig S7A

Fig S7B

Rebuttal figure 2 (A) The expression and variation of miR-5p and miR-3p in gastric cancer tissues. (B) The expression change of miR-5p and miR-3p from the same miRNA precursor in gastric cancer tissues compared with the normal controls.

miR-574 is indicated by arrows.

References

1. Wang X, Lu X, Geng Z, Yang G, Shi Y. LncRNA PTCSC3/miR-574-5p Governs Cell Proliferation and Migration of Papillary Thyroid Carcinoma via Wnt/ β -Catenin Signaling. *J Cell Biochem.* 2017 Dec;118(12):4745-4752.
2. Zhou R, Zhou X, Yin Z, Guo J, Hu T, Jiang S, Liu L, Dong X, Zhang S, Wu G. MicroRNA-574-5p promotes metastasis of non-small cell lung cancer by targeting PTPRU. *Sci Rep.* 2016 Oct 20;6:35714.
3. Ma DL, Li JY, Liu YE, Liu CM, Li J, Lin GZ, Yan J. Influence of continuous intervention on growth and metastasis of human cervical cancer cells and

- expression of RNA miR-574-5p. *J Biol Regul Homeost Agents*. 2016 Jan-Mar;30(1):91-102. PubMed PMID: 27049079.
4. Zhou R, Zhou X, Yin Z, Guo J, Hu T, Jiang S, Liu L, Dong X, Zhang S, Wu G. Tumor invasion and metastasis regulated by microRNA-184 and microRNA-574-5p in small-cell lung cancer. *Oncotarget*. 2015 Dec 29;6(42):44609-22.
 5. Cui Z, Tang J, Chen J, Wang Z. Hsa-miR-574-5p negatively regulates MACC-1 expression to suppress colorectal cancer liver metastasis. *Cancer Cell Int*. 2014 Jun 7;14:47.
 6. Ji S, Ye G, Zhang J, Wang L, Wang T, Wang Z, Zhang T, Wang G, Guo Z, Luo Y, Cai J, Yang JY. miR-574-5p negatively regulates Qki6/7 to impact β -catenin/Wnt signalling and the development of colorectal cancer. *Gut*. 2013 May;62(5):716-26.
 7. Yao P, Wu J, Lindner D, Fox PL. Interplay between miR-574-3p and hnRNP L regulates VEGFA mRNA translation and tumorigenesis. *Nucleic Acids Res*. 2017 Jul 27;45(13):7950-7964.
 8. Ujihira T, Ikeda K, Suzuki T, Yamaga R, Sato W, Horie-Inoue K, Shigekawa T, Osaki A, Saeki T, Okamoto K, Takeda S, Inoue S. MicroRNA-574-3p, identified by microRNA library-based functional screening, modulates tamoxifen response in breast cancer. *Sci Rep*. 2015 Jan 6;5:7641.
 9. Xu H, Liu X, Zhou J, Chen X, Zhao J. miR-574-3p acts as a tumor promoter in osteosarcoma by targeting SMAD4 signaling pathway. *Oncol Lett*. 2016 Dec;12(6):5247-5253.
 10. Chiyomaru T, Yamamura S, Fukuhara S, Hidaka H, Majid S, Saini S, Arora S, Deng G, Shahryari V, Chang I, Tanaka Y, Tabatabai ZL, Enokida H, Seki N, Nakagawa M, Dahiya R. Genistein up-regulates tumor suppressor microRNA-574-3p in prostate cancer. *PLoS One*. 2013;8(3):e58929.

The associations of miR-574-5p and -3p in GC development as well as the functional characterization and target gene analysis are solid, however, the claim that highly complementary targets of miR-574-5p/-3p can induce miRNA decay and mediate

arm-imbalance of miR-574-5p/-3p in GC progression relies entirely on associations and is not supported by sufficient data. The fact that overexpression of artificial sponges can lead to degradation of mature miRNAs is well known. What the authors need to convincingly show is that the expression levels of targets/ the number of miRNA binding sites in IBA57-AS1, KLRC2, S100A1 and TMEM54) reach levels in GC that are sufficient to induce miRNA decay. This cannot be demonstrated by relative expression studies.

Reply: We acknowledged that the original claim “highly complementary targets of miR-574-5p/-3p can induce miRNA decay and mediate arm-imbalance of miR-574-5p/-3p in GC progression” is not convincing with the association study in the primary submitted manuscript. We therefore detected the absolute copy number of the four targets (IBA57-AS1, KLRC2, S100A1 and TMEM54) and miR-574-5p/-3p in gastric cancer patients with inversely differential expression of miR-574-5p/-3p. The absolute RNA copy number was determined by absolute quantitative real-time PCR, which was based on a standard curve established with corresponding purified PCR products or plasmids. The results showed that the absolute copy number of these targets was massive compared to miRNAs, suggesting the expression levels of these targets could reach the level that were sufficient to induce miRNA decay (Rebuttal figure 3). Meanwhile, we also observed that miR-574-5p targets (IBA57-AS1, KLRC2) were obviously down-regulated, while miR-574-3p targets (S100A1 and TMEM54) were up-regulated in these patients, which is in contrast to the corresponding miRNA change, supporting the potential miRNA decay might be induced by these targets in gastric cancer patients. On the other hand, the number of the highly complementary miRNA binding sites in IBA57-AS1, KLRC2, S100A1 and TMEM54 predicted by RNA Hybrid was presented in Rebuttal figure 4, however the real miRNA binding sites in these targets could not be determined.

Please refer to the revised Fig 5 for related information.

Fig 5F

Rebuttal figure 3 The absolute RNA copy number of the four targets (IBA57-AS1, KLRC2, S100A1 and TMEM54) and miR-574-5p/-3p in gastric cancer patients with previously identified inversely differential expression of miR-574-5p/-3p.

Figure 5A

Rebuttal figure 4 Schematic representation of the four targets, including the miR-574 binding sites and cloning fragment for overexpression.

Does silencing of IBA57-AS1, KLRC2, S100A1 and TMEM54 in GC cells result in a global repression of 574-5p targets?

Reply: We thank this reviewer for this important suggestion. As required, we interfered the expression of IBA57-AS1, KLRC2, S100A1 and TMEM54 in MGC-803 cells and performed RNA-sequencing to uncover the global change of the targets of miR-574-5p/3p. As the basic expression level of IBA57-AS1, KLRC2 in MGC-803 cells was relatively low according to our RNA-sequencing data, we overexpressed IBA57-AS1 and KLRC2 instead of suppressed them. As a result, when IBA57-AS1, KLRC2 was overexpressed (Rebuttal figure 5), the potential targets of miR-574-5p, defined as highly enriched in miR-574-5p pulled down RNAs and decreased with miR-574-5p overexpression (indicated as blue dots in Rebuttal figure 6 left), were up-regulated when compared with all genes ($p=0.012$, $p=0.0099$) (Rebuttal figure 7). When S100A1 and TMEM54 were silenced with siRNAs (Rebuttal figure 5), the potential targets of miR-574-3p, defined as highly enriched in miR-574-3p pulled down RNAs and decreased with miR-574-3p overexpression (indicated as blue dots in Rebuttal figure 6 right), were indeed repressed obviously when compared with all genes ($p=0.0087$, $p=0.0071$) (Rebuttal figure 7). Meanwhile, the expression of partial of the most potential targets (enrichment ratio >1.5 , fold change >1.5 from Rebuttal figure 6) was also validated with quantitative PCR. The results indicated that 61% (25/41) and 66% (27/41) of the potential miR-574-5p targets were increased with IBA57-AS1, KLRC2 overexpression respectively. 87% (33/38) and 79% (30/38) of the potential miR-574-3p targets were inhibited by knocking down S100A1 and TMEM54, respectively (Rebuttal figure 8). These results all showed that interfering the four highly complementary targets could induce global change of miR-574 targets, which indirectly supported the notion that alteration of miR-574 was induced by the non-canonical highly complementary targets.

Please refer to the revised Fig 5 and Fig S4 for related information.

Fig S4A

Fig S4C

Rebuttal figure 5 (A) IBA57-AS1 and KLRC2 were overexpressed in MGC-803 cells. (C) S100A1 and TMEM54 were suppressed by siRNAs in MGC-803 cells.

Fig 4D

Rebuttal figure 6 The expression change of miR-574-5p/-3p pulled down targets in miR-574-5p/-3p overexpressed cells. ER: enrichment ratio.

Fig 5E

Rebuttal figure 7 Global expression of the potential miR-574 targets in MGC-803 cells with IBA57-AS1, KLRC2 overexpression and S100A1, TMEM54 knock down respectively.

Fig 4E

Fig 4F

Rebuttal figure 8 (E) The expression change of the potential miR-574-5p targets

when IBA57-AS1, KLRC2 was overexpressed. (F) The expression change of the potential miR-574-3p targets when S100A1, TMEM54 was suppressed.

To infer a mechanism of miRNA decay simply because miR-574-5p level were decreased without a concomitant decrease in primary-miR-574 transcripts is flawed, since other mechanisms may account for this. Did the authors notice tailing and trimming in their sequencing data or in Northern blots?

Reply: This is a very constructive question. As the previous RNA-sequencing data was not including the small RNAs, we performed small RNA sequencing and analysis in MGC-803 cells with IBA57-AS1, KLRC2, S100A1 and TMEM54 overexpression (Rebuttal figure 9). The total miR-574-5p reads decreased dramatically when IBA57-AS1, KLRC2 was overexpressed, while proportion of the tailing and trimming miR-574-5p was increased apparently, especially the trimming ones, suggesting the miRNA decay induced by the two targets (Rebuttal figure 10). In line with this finding, when S100A1 or TMEM54 was overexpressed, total miR-574-3p reads declined accompanied by slightly up-regulation of the tailing and trimming miR-574-3p (Rebuttal figure 10). Therefore, as the reviewer suggested, the small RNA sequencing data together with the unchanged expression of primary-miR-574 transcripts indicated that the non-canonical highly complementary targets indeed induced miRNA decay.

Please refer to the revised Fig 5 and Fig S4 for related information.

Fig S4

Rebuttal figure 9 (A) IBA57-AS1, KLRC2, S100A1 and TMEM54 were overexpressed in MGC-803 cells.

Fig 5D

Rebuttal figure 10 Left panel: Bar plot indicating the absolute miRNA expression levels in MGC-803 cells with overexpression of the targets. Middle panel: the proportion of different miRNA isoforms. Right panel: isoform sequences displayed by color-coding.

Hyberbels in the text should be avoided throughout the paper

Reply: We thank the reviewer for this detail comment. We have tried our best to avoid hyberbels throughout the paper.

Reviewer #2, Expertise: gastric cancer, miRNA

(Remarks to the Author):

This interesting manuscript demonstrates the role miR-574-5p and miR-574-3p in gastric cancer. Manuscript is quite extensive. The authors provided a number of lines of evidence demonstrating the opposite functions of miR-574-5p and miR-574-3p in gastric cancer as well as the critical function of the ratio of these two molecules. I believe that is the first or one of the first papers demonstrating that the ratio of two microRNAs derived from the same precursor is important in carcinogenesis.

1. It is not clear why the authors started with isoforms of miR-219, miR-369 and miR-574. Did they do any preliminary screening?

Reply: This is a similar concern to the first question of reviewer 1.

Actually, the current study of the role of miR-574 in gastric cancer was initiated from another project which investigated promoter methylation level change in gastric cancer based on our previous MeDIP (Methylated DNA immunoprecipitation) combined with a human miRNA tiling microarray analysis (Rebuttal Figure 1A). These results were not included in the current manuscript as it will be reported in another work, which mainly focused on the role of miR-369 in gastric cancer (Molecular Oncology, MOLONC-18-0804, under revision).

Briefly, we identified several potential miRNAs correlated with hypermethylation and up-regulated upon 5-Azacytidine (AZA) treatment in gastric cancer, which were miR-10a/b, miR-33b, miR-219, miR-369, miR-574 and so on (Rebuttal Figure 1B, C). Furthermore, the divergent role of miR-574-5p and miR-574-3p in various cancers was reported (Ref 1-10) as described in the Introduction section, which drove us curiosity to uncover the detail function of two arms of miR-574 in gastric cancer. We then examined their expression in many gastric cancer tissues compared with the adjacent non-cancerous tissues, as well as the expression of the two arms of miR-219 and miR-369.

Rebuttal figure 1 (A) Schematic procedure of DNA methylation-induced miRNA identification. (B) The correlation between specific miRNAs and hypermethylated regions. Red ellipses indicated the hypermethylated regions. Green ellipses indicated the hypomethylated regions. Gray ellipses indicated no significant difference. The red arrow indicated miR-574. (C) AZA induced the expression of 13 miRNAs in a concentration-dependent manner in gastric cancer cells as shown by qRT-PCR analysis. The red arrow indicated miR-574-3p.

To further solve the reviewer’s concern, we performed a comprehensive expression analysis of two arms of a miRNA (miR-5p and miR-3p) in gastric cancer tissues using the entire TCGA gastric cancer database (424 patients) combined with another miRNA profiling dataset which included 73 gastric cancer patients and 94 normal controls. 1749 miRNAs were included in this analysis. Simultaneous detection of both arms was found in 838 miRNAs. Among them, 301 (35.91%) miRNA pairs exhibited same expression tendencies in gastric cancer compared with normal controls, while 76 (9.07%) miRNA pairs including miR-574 showed opposite expression alternation

(Rebuttal figure 2A). Thus, we can tell that miR-574 is not the only miRNA with which a miRNA arm-selection was observed, and 76 miRNAs showed an arm ratio variation in gastric cancer. Please refer to the heatmap presented in Rebuttal figure 2B and supplementary table 9. Anyway, the underlying mechanism of the dynamic miRNA arm-preference in gastric cancer remains unclear, but we believe the target RNA directed miRNA degradation (TDMD) demonstrated in the current manuscript might be one of them.

Finally, we rephrased the “Result” for selecting miR-574 as our research interest and the supplemented miRNA global analysis have been added in the revised manuscript.

Please refer to the revised Fig S7 and supplementary table 9 for related information.

Rebuttal figure 2 (A) The expression and variation of miR-5p and miR-3p in gastric

cancer tissues. (B) The expression change of miR-5p and miR-3p from the same miRNA precursor in gastric cancer tissues compared with the normal controls.

miR-574 is indicated by arrows.

References

1. Wang X, Lu X, Geng Z, Yang G, Shi Y. LncRNA PTCSC3/miR-574-5p Governs Cell Proliferation and Migration of Papillary Thyroid Carcinoma via Wnt/ β -Catenin Signaling. *J Cell Biochem.* 2017 Dec;118(12):4745-4752.
2. Zhou R, Zhou X, Yin Z, Guo J, Hu T, Jiang S, Liu L, Dong X, Zhang S, Wu G. MicroRNA-574-5p promotes metastasis of non-small cell lung cancer by targeting PTPRU. *Sci Rep.* 2016 Oct 20;6:35714.
3. Ma DL, Li JY, Liu YE, Liu CM, Li J, Lin GZ, Yan J. Influence of continuous intervention on growth and metastasis of human cervical cancer cells and expression of RNA miR-574-5p. *J Biol Regul Homeost Agents.* 2016 Jan-Mar;30(1):91-102. PubMed PMID: 27049079.
4. Zhou R, Zhou X, Yin Z, Guo J, Hu T, Jiang S, Liu L, Dong X, Zhang S, Wu G. Tumor invasion and metastasis regulated by microRNA-184 and microRNA-574-5p in small-cell lung cancer. *Oncotarget.* 2015 Dec 29;6(42):44609-22.
5. Cui Z, Tang J, Chen J, Wang Z. Hsa-miR-574-5p negatively regulates MACC-1 expression to suppress colorectal cancer liver metastasis. *Cancer Cell Int.* 2014 Jun 7;14:47.
6. Ji S, Ye G, Zhang J, Wang L, Wang T, Wang Z, Zhang T, Wang G, Guo Z, Luo Y, Cai J, Yang JY. miR-574-5p negatively regulates Qki6/7 to impact β -catenin/Wnt signalling and the development of colorectal cancer. *Gut.* 2013 May;62(5):716-26.
7. Yao P, Wu J, Lindner D, Fox PL. Interplay between miR-574-3p and hnRNP L regulates VEGFA mRNA translation and tumorigenesis. *Nucleic Acids Res.* 2017 Jul 27;45(13):7950-7964.
8. Ujihira T, Ikeda K, Suzuki T, Yamaga R, Sato W, Horie-Inoue K, Shigekawa T, Osaki A, Saeki T, Okamoto K, Takeda S, Inoue S. MicroRNA-574-3p, identified by

microRNA library-based functional screening, modulates tamoxifen response in breast cancer. *Sci Rep.* 2015 Jan 6;5:7641.

9. Xu H, Liu X, Zhou J, Chen X, Zhao J. miR-574-3p acts as a tumor promoter in osteosarcoma by targeting SMAD4 signaling pathway. *Oncol Lett.* 2016 Dec;12(6):5247-5253.
10. Chiyomaru T, Yamamura S, Fukuhara S, Hidaka H, Majid S, Saini S, Arora S, Deng G, Shahryari V, Chang I, Tanaka Y, Tabatabai ZL, Enokida H, Seki N, Nakagawa M, Dahiya R. Genistein up-regulates tumor suppressor microRNA-574-3p in prostate cancer. *PLoS One.* 2013;8(3):e58929.

2. First the authors identified 2 targets QKI6 and ACVR1B, and show their effect in gastric cancer. Then they did systematic identification a number of additional targets. If, QKI6 and ACVR1B account for all the effects of miR-574-5p and miR-574-3p as manuscript suggests, what is the reason then to speculate about roles of additional targets?

Reply: We thank the reviewer for this comment. The aim to identify additional targets was to investigate the regulatory network of miR-574-5p/-3p, particularly to find the inducers for the opposite changes of miR-574-5p/-3p arm in gastric carcinogenesis, since miRNA targets have been reported to regulate miRNA expression through target-mediated miRNA protection (TMMP) or target RNA directed miRNA degradation (TDMD).

By pulling down RNAs of the biotinylated miR-574-5p/-3p, we indeed detected non-canonical highly complementary targets which can regulate miRNA degradation, this is also supported by our newly added small RNA sequencing data as we described above (Rebuttal figure 10).

Please refer to the revised Fig 5 for related information.

Fig 5D

Rebuttal figure 10 Left panel: Bar plot indicating the absolute miRNA expression levels in MGC-803 cells with overexpression of the targets. Middle panel: the proportion of different miRNA isoforms. Right panel: isoform sequences displayed by color-coding.

We performed small RNA sequencing and analysis in MGC-803 cells with IBA57-AS1, KLRC2, S100A1 and TMEM54 overexpression (Rebuttal figure 9). The total miR-574-5p reads decreased dramatically when IBA57-AS1, KLRC2 was overexpressed, while proportion of the tailing and trimming miR-574-5p was increased apparently, especially the trimming ones, suggesting the miRNA decay induced by the two targets (Rebuttal figure 10). In line with this finding, when S100A1 or TMEM54 was overexpressed, total miR-574-3p reads declined accompanied by slightly up-regulation of the tailing and trimming miR-574-3p (Rebuttal figure 10). Therefore, as the reviewer suggested, the small RNA sequencing data together with the unchanged expression of primary-miR-574 transcripts

indicated that the non-canonical highly complementary targets indeed induced miRNA decay.

Although our data revealed that QKI6 and ACVR1B were the key targets of miR-574-5p/3p in gastric cancer, we also observed other miRNAs were pulled down by miR-574-5p/-3p and decreased with their overexpression. miRNA often binds and interacts with multiple targets to exert their function, we speculated these genes might also be miR-574-5p/-3p potential targets in gastric cancer cells, which of course requires further experimental validation. The exploration of these targets suggested that miR-574-5p/-3p acted in gastric carcinogenesis via coordinately targeting multiple associated genes not only QKI6 or ACVR1B. Indeed the overexpression of QKI6 or ACVR1B almost completely rescued the effects of miR-574-5p/-3p, but in some experiments, e.g. the gastric cancer cell invasion assay, the phenotypes were not fully rescued by QKI6 or ACVR1B overexpression, in addition, the alteration of global gene expression changes was not examined. Therefore, the aim of these studies was to uncover the regulatory networks of miR-574-5p/-3p and their additional targets in gastric cancer cells. It is true that the functions of these additional targets was not closely linked to the main topic and was therefore moved to supplementary figures so we could focus more on the contribution of the non-canonical highly complementary targets to regulate miR-574-5p/-3p decay. Thus, we changed “Systematic identification of miR-574-5p/-3p targets in GC cells” to “Regulatory network of miR-574-5p/-3p targets in GC cells” in the revised manuscript.

3. How many targets of miR-574-5p and miR-574-3p were higher on Targetscan list than ones they studied, and why the authors focused on QKI6 and ACVR1B?

Reply: This is a reasonable concern. It is reported that Qki6/7 can be negatively regulated by miR-574-5p directly in colorectal cancer (Ref 11), we therefore questioned whether they were also direct targets of miR-574-5p in gastric cancer. The dual luciferase reporter assay and western blot confirmed miR-574-5p can

directly target Qki6 in gastric cancer, but Qki7 was not detected in gastric cancer cells. As suggested by the reviewer, we obtained 15 direct potential targets via Targetscan and Miranda with considering their reported roles in cancer. Although dual luciferase reporter assay indicated that 3'UTR region of 5 out of 15 targets were negatively regulated by miR-574-3p, the protein immunoblot assay only suggested ACVR1B protein was significantly decreased by miR-574-3p and thus identified as miR-574-3p target in gastric cancer. Although ACVR1B was not ranked top in the Targetscan prediction list, it was proved to be a real target of miR-574-3p in gastric cancer. Therefore, we only focused on QKI6 and ACVR1B for the current study.

References

11. Ji S, Ye G, Zhang J, Wang L, Wang T, Wang Z, Zhang T, Wang G, Guo Z, Luo Y, Cai J, Yang JY. miR-574-5p negatively regulates Qki6/7 to impact β -catenin/Wnt signalling and the development of colorectal cancer. *Gut*. 2013 May;62(5):716-26.

4. There are over 100 publications about miR-574-5p and miR-574-3p, many of them about their role in cancer, and show that multiple other important targets of these two microRNAs are important in cancer (authors cited some of these papers). Are these targets expressed in gastric cancer, particularly vs QKI6 and ACVR1B? Are they inhibited by these two microRNAs? Were they identified by the authors in their systematic approach?

Reply: We appreciate this instructive suggestion. After carefully exploration of these relevant papers, we found out 9 studies reported miR-574-5p targets (SCAI, PTPRU, EPAS1, MACC1, ZEB1, ZDHHC14, REL, CBR1, FOXN3, CerS1) and 8 reported miR-574-3p targets (ZEB1, CLTC, SMAD4, RAC1, EGFR, EP300, ERH, CUL2) in different cancers. We further analyzed the RNA level of these targets in our RNA-sequencing data of MGC-803 cells with miR-574-5p/3p overexpression. All these genes except one miR-574-5p target MACC1, were expressed in MGC-803 cells. Four out of eight miR-574-5p targets and 3 out of miR-574-3p targets were down-regulated with miR-574-5p/3p overexpression. In addition, according to our systematic miRNA pull

down data, 7 out of 8 miR-574-5p targets and 2 out of 8 miR-574-3p targets were clearly enriched in miR-574-5p and miR-574-3p pulled down RNAs, respectively (Rebuttal figure 11). These results suggested that the inconsistency of reported miR-574-5p/3p targets between other cancers and gastric cancer, which might due to the tissue specificity and dynamic accessibility of the miRNA binding sites.

Please refer to the revised Fig S3 for related information.

Fig S3C

Rebuttal figure 11 (C) The expression change of the reported miR-574 targets in MGC-803 cells with miR-574 overexpression; The relative enrichment of the reported miR-574 targets by biotinylated miR-574 mimics in gastric cancer cells.

5. Previous studies identified QKI6 as a target of miR-574-5p (as mentioned by the authors). So these experiments including QKI6 are rather a confirmation of previous data in different cancer.

Reply: We admitted this part of the study might lack of novelty. It is true that QKI6 results were indeed a confirmation of previous data, but we think it might be also used as an “internal control” to increase our study’s reliability. As we reported, we

also identified other novel targets of miR-574-5p in gastric cancer, as revealed by the microRNA pull down assay. However, the major interest for the current study is to seek the mechanism that why the two arms of the same precursor were reversely expressed in gastric cancer tissues, so we did not make major efforts to further investigate these novel targets of miR-574-5p in gastric cancer, which is of course one of our future directions.

6. Introduction and discussion sections need to be shortened.

Reply: We rephrased the introduction and discussion sections as the reviewer suggested.

Reviewers' comments:

Reviewer #2 (Remarks to the Author):

My comments were addressed.

Reviewer #3 (Remarks to the Author):

This revised manuscript has been significantly improved with regard to concerns from Reviewer #1. All points have been addressed, and convincing new experiments have been performed and new data provided in most cases.

For the initial concern that there is not sufficient evidence supporting a target-induced miRNA degradation, the authors have provided several new pieces of data. One of the most critical data pieces is that the knockdown of miR-574-3p targets S100A1 or TMEM54 led to a significant enrichment of predicted miR-574-3p targets among downregulated genes (Fig 5E). The reason this experiment is important is because all other experiments supporting this target-mediated miRNA decay model is by over-expressing targets, rather than reducing endogenous targets. However, the authors did not show whether knocking down S100A1 or TMEM54 affected the levels of miR-574-3p. Without examining miR-574-3p levels, Fig 5E results may be explained by the competing endogenous RNA (ceRNA) theory, which states that reducing endogenous miRNA targets may lead to more effective targeting of other targets through target competition rather than miRNA decay.

Another piece of evidence supporting a decay model, which is Fig 5D, is missing statistical analysis to support a significant shift to trimmed miRNA molecules.

In addition to the concerns raised by the first round of review, one thing that is helpful, although not critical, is to examine the expression of miR-574-5p when modulating the level of miR-574-3p, and vice versa. I am somewhat surprised that the authors did not show data on this front, given the 5p and 3p strands are quite complementary (Fig 1A), and may result in mutual regulation through AGO2-mediated cutting and degradation. If true, it will support a mutual regulatory relationship between 574-3p and 5p miRNAs, and help explain the opposite expression trends, rather than a simple target-mediated decay model.

Lastly, on a minor note, for Fig 6A, it will be helpful to show statistical significance.

Reviewers' comments:

Reviewer #3 (Remarks to the Author):

This revised manuscript has been significantly improved with regard to concerns from Reviewer #1. All points have been addressed, and convincing new experiments have been performed and new data provided in most cases.

For the initial concern that there is not sufficient evidence supporting a target-induced miRNA degradation, the authors have provided several new pieces of data. One of the most critical data pieces is that the knockdown of miR-574-3p targets S100A1 or TMEM54 led to a significant enrichment of predicted miR-574-3p targets among downregulated genes (Fig 5E). The reason this experiment is important is because all other experiments supporting this target-mediated miRNA decay model is by over-expressing targets, rather than reducing endogenous targets. However, the authors did not show whether knocking down S100A1 or TMEM54 affected the levels of miR-574-3p. Without examining miR-574-3p levels, Fig 5E results may be explained by the competing endogenous RNA (ceRNA) theory, which states that reducing endogenous miRNA targets may lead to more effective targeting of other targets through target competition rather than miRNA decay.

Reply: We appreciate the helpful comments from the reviewer. We examined the expression level of miR-574-3p in MGC-803 cells when S100A1 or TMEM54 was knocked down. As shown in FigS4C, when S100A1 or TMEM54 was successfully suppressed, miR-574-3p indeed increase statistically significantly but not impressively. This result together with the decline and trimming of miR-574-3p with S100A1 or TMEM54 overexpression (Fig 5B, D) indicated that endogenous S100A1 or TMEM54 can really induce miR-574-3p decay. The unremarkable change of miR-574-3p with S100A1 or TMEM54 knock down raised that it was possible that not all of S100A1 or TMEM54 mRNA were bound by miR-574-3p. Considering the speculation from the reviewer, we deduced that more effective targeting of other miR-574-3p targets together with reduction of miR-574-3p decay might simultaneously contribute to the down-regulation of other targets of miR-574-3p with S100A1 or TMEM54 knock-down. We discussed it in the revised manuscript.

Supplementary Figure4 (C) Examination of miR-574-3p expression in MGC-803 cells when S100A1

and TMEM54 were suppressed.

Figure 5 (B) Relative expression of miR-574-5p/-3p in GC cells with overexpression of IBA57-AS1, KLRC2 or S100A1, TMEM54.

Figure 5 (D) Left panel: Bar plot indicating the absolute miRNA expression levels in MGC-803 cells with overexpression of the targets. Middle panel: the proportion of different miRNA isoforms. Right panel: isoform sequences displayed by color-coding.

Another piece of evidence supporting a decay model, which is Fig 5D, is missing statistical analysis to support a significant shift to trimmed miRNA molecules.

Reply: We appreciate the reasonable comments from the reviewer. In fact, we conducted small RNA sequencing in two replicates but calculated an average value previously. According to the suggestions from the reviewer, we reanalyzed the small RNA sequencing data and the statistical differences were indicated. The data was also supplemented in Table S7. As shown in Fig 5D, the decrease of miR-574-5p/3p total reads, the decrease of annotated mature miR-574-5p/3p, as well as the up-regulation of trimming miR-574-5p/3p were statistically significant. This result strongly indicated a significant shift to trimmed miRNA molecules when the highly

complementary targets were overexpressed.

Figure 5 (D) Left panel: Bar plot indicating the absolute miRNA expression levels in MGC-803 cells with overexpression of the targets. Middle panel: the proportion of different miRNA isoforms. Right panel: isoform sequences displayed by color-coding.

In addition to the concerns raised by the first round of review, one thing that is helpful, although not critical, is to examine the expression of miR-574-5p when modulating the level of miR-574-3p, and vice versa. I am somewhat surprised that the authors did not show data on this front, given the 5p and 3p strands are quite complementary (Fig 1A), and may result in mutual regulation through AGO2-mediated cutting and degradation. If true, it will support a mutual regulatory relationship between 574-3p and 5p miRNAs, and help explain the opposite expression trends, rather than a simple target-mediated decay model.

Reply: We appreciate the instructive suggestions from the reviewer. We examined the expression of miR-574-5p when miR-574-3p was overexpressed or suppressed, and also examined the expression of miR-574-3p when miR-574-5p was modulated. As shown in FigS4G, when miR-574-5p was successfully overexpressed or suppressed, miR-574-3p was not changed. Similarly, when miR-574-3p was successfully overexpressed or inhibited, miR-574-5p was not changed too, indicating that partially complementary arms of miR-574 cannot induce mature miRNA decay. To date, the particular mechanism of highly complementary targets inducing microRNA decay (TDMD) was largely unknown. Terminal-Uridylyl-Transferase TUT1 and 3'-5' exoribonuclease DIS3L2 have been demonstrated to interact with Argonaute2 to participate in TDMD [1]. Besides, extended complementary to the 3'-end of miRNA was critical for induction of TDMD [2, 3]. The 3' end mismatch between two miR-574 arms and the existing bulge in the 3' end of miR-574-3p might not trigger TDMD efficiently (Fig 1A). The specific molecular details and the exact complementary rule triggering TDMD needs to be explored further. Here, we can only conclude that the opposite expression of miR-574-5p/3p in gastric cancer was not due to the

mutual regulation of the two arms but the dynamic expression of their highly complementary targets in gastric cancer according to our results.

Supplementary Figure4 (G) Examination of miR-574-5p and miR-574-3p expression in MGC-803 cells when miR-574-5p/3p was overexpressed or suppressed.

Figure1 (A) The predicted secondary structure of miR-219, miR-369 and miR-574 precursors.

References

1. Haas G, Cetin S, Messmer M, Chane-Woon-Ming B, Terenzi O, Chicher J, Kuhn L, Hammann P, Pfeffer S. Identification of factors involved in target RNA-directed microRNA degradation. *Nucleic Acids Res.* 2016 Apr 7;44(6):2873-87.
2. Ameres SL, Horwich MD, Hung JH, Xu J, Ghildiyal M, Weng Z, Zamore PD. Target RNA-directed trimming and tailing of small silencing RNAs. *Science.* 2010 Jun 18;328(5985):1534-9.

3. de la Mata M, Gaidatzis D, Vitanescu M, Stadler MB, Wentzel C, Scheiffele P, Filipowicz W, Großhans H. Potent degradation of neuronal miRNAs induced by highly complementary targets. EMBO Rep. 2015 Apr;16(4):500-11.

Lastly, on a minor note, for Fig 6A, it will be helpful to show statistical significance.

Reply: We appreciate the reasonable comments from the reviewer. The statistical significance was analyzed and indicated in revised Fig 6A.

Figure 6 (A) The proliferation rate of MGC-803 cells, treated with miR-574-5p/3p mimics (5p-OE/3p-OE) and miR-574-5p/3p inhibitors (5p-KD/3p-KD) at multiple combinations.

REVIEWERS' COMMENTS:

Reviewer #3 (Remarks to the Author):

The authors have shown new data to address all of my concerns, and I appreciate their efforts. Overall, I think their study is suitable for Nature Communications. However, I do have two issues with the revision, that can, hopefully, be quickly resolved by changes in text and graphs/statistics.

1. One of the most important experiments to support the target-mediated miRNA decay model is shown in Figure 5E, as well as the newly added Supp Figure 4C. This is the only experiment in the whole manuscript to prove that endogenous levels of highly complementary targets of miR-574-3p can lead to miRNA expression changes. However, as commented by the authors in their rebuttal, the changes of miR-574-3p upon efficient knockdown of miR-574-3p targets (Supp Fig 4C) are not strong, despite being statistically significant. Accordingly, I am not fully convinced that target-mediated decay is the main mechanism by which miR-574-5p/3p ratios are changed in GC, although I do agree that there is sufficient data to support a partial contribution of this mechanism in their system. With this said, the functions of miR-574-5p and 3p in GC are very well demonstrated in the manuscript and is worth publishing.

What I recommend the authors to do is to avoid over-claims in their title and abstract, as well as throughout the manuscript, on the contribution of target-mediated decay to the miRNA arm imbalance. For instance, title may be "microRNA arm-imbalance promotes gastric cancer progression" or "microRNA arm-imbalance promotes gastric cancer progression regulated partially by target-mediated decay". Similarly, emphasis of partial contribution would be important in abstract and main text.

2. The modified Figure 5D has only two biological samples, according to author's rebuttal (which is not clearly stated in the corresponding figure legend). So it is not appropriate to plot error bars and use t-test for statistical evaluation. It probably makes more sense to perform Fisher Exact test by using the absolute read count of specific isoforms and utilize the number of total reads minus that of a specific isoform as input.

Response to reviewer

Reviewer #3 (Remarks to the Author):

The authors have shown new data to address all of my concerns, and I appreciate their efforts. Overall, I think their study is suitable for Nature Communications. However, I do have two issues with the revision, that can, hopefully, be quickly resolved by changes in text and graphs/statistics.

1. One of the most important experiments to support the target-mediated miRNA decay model is shown in Figure 5E, as well as the newly added Supp Figure 4C. This is the only experiment in the whole manuscript to prove that endogenous levels of highly complementary targets of miR-574-3p can lead to miRNA expression changes. However, as commented by the authors in their rebuttal, the changes of miR-574-3p upon efficient knockdown of miR-574-3p targets (Supp Fig 4C) are not strong, despite being statistically significant. Accordingly, I am not fully convinced that target-mediated decay is the main mechanism by which miR-574-5p/3p ratios are changed in GC, although I do agree that there is sufficient data to support a partial contribution of this mechanism in their system. With this said, the functions of miR-574-5p and 3p in GC are very well demonstrated in the manuscript and is worth publishing.

What I recommend the authors to do is to avoid over-claims in their title and abstract, as well as throughout the manuscript, on the contribution of target-mediated decay to the miRNA arm imbalance. For instance, title may be “microRNA arm-imbalance promotes gastric cancer progression” or “microRNA arm-imbalance promotes gastric cancer progression regulated partially by target-mediated decay”. Similarly, emphasis of partial contribution would be important in abstract and main text.

Reply: We appreciated the helpful suggestions from the reviewer. We emphasized the partial contribution of targets mediated decay to the miR-574 arm imbalance throughout

the manuscript. All changes about this were marked as red in the revised manuscript. Title was changed to “microRNA arm-imbalance partially from targets-mediated decay promotes gastric cancer progression” according to the reviewer’s suggestion.

2. The modified Figure 5D has only two biological samples, according to author’s rebuttal (which is not clearly stated in the corresponding figure legend). So it is not appropriate to plot error bars and use t-test for statistical evaluation. It probably makes more sense to perform Fisher Exact test by using the absolute read count of specific isoforms and utilize the number of total reads minus that of a specific isoform as input.

Reply: We appreciated the helpful suggestions from the reviewer. We performed Fisher Exact test as the reviewer suggested. The results clearly showed that the decrease of miR-574-5p/3p total reads, the decrease of annotated mature miR-574-5p/3p, as well as the up-regulation of trimming miR-574-5p/3p were statistically significant. According to editorial requests, the corresponding data points were indicated as dot plots in the bar charts. The two replicates of miRNA sequencing data and the statistical methods used here were added to the figure legends. P value of Fisher Exact test was also provided in supplementary table 7.

Figure5 (D) Left panel: Bar plot indicating the absolute miRNA expression levels in MGC-803 cells with overexpression of the targets. Middle panel: the proportion of different miRNA isoforms. Right panel: isoform sequences displayed by color-coding. miRNA sequencing was performed in two replicates and Fisher Exact test was used. *P<0.05, **P<0.01, ***P<0.001,

miR-574-5p isoforms	P value (IBA57-AS1 vs Con)	P value (KLRC2 vs Con)
TGAGTGTGTGTGTGTGAGTGTGTGTC	NA	NA
TGAGTGTGTGTGTGTGAGTGTGTGT	NA	NA
TGAGTGTGTGTGTGTGAGTGTGTG	0.0018	0.0050
TGAGTGTGTGTGTGTGAGTGTGT	0.0024	0.0009
TGAGTGTGTGTGTGTGAGTGTG	0.0003	0.0320
TGAGTGTGTGTGTGTGAGTGT	0.7226	0.0069
TGAGTGTGTGTGTGTGAGTG	0.0200	0.0690
TGAGTGTGTGTGTGTGAGT	NA	0.1817

TGAGTGTGTGTGTGTGAG	0.1817	0.1817
total	0.0029	0.0056
miR-574-3p isoforms	P value (S100A1 vs Con)	P value (TMEM54 vs Con)
CACGCTCATGCACACACCCACAC	0.1140	0.0360
CACGCTCATGCACACACCCACA	0.0050	0.0171
CACGCTCATGCACACACCCAC	0.0055	0.0122
CACGCTCATGCACACACCCA	0.0410	0.0110
CACGCTCATGCACACACCC	0.0043	0.0355
CACGCTCATGCACACACC	0.3349	0.4680
total	0.0108	0.0156

Supplementary table 7 P value of Fisher Exact test of miR-574-5p/-3p isoforms with IBA57-AS1, KLRC2, S100A1 and TMEM54 overexpression.